# Magnetization generation and giant nonlinear transport at symmetry-engineered interfaces

Hang-Bo Zhang [1,7], Zhen-Yu Ding [2,7], Yi-Ning Xie [3], Zheng-Hao Li[1,4], Eoin Moynihan[3], Ana M. Sanchez [3], WenGuang Zhu [1,2,5], Yang Gao [1,2,5] ✉, Yoshihiro Iwasa [6], Marin Alexe [3] & Ming-Min Yang [1,4] ✉

Interfaces in heterostructures possess inherent inversion asymmetry and display diverse physical effects, however, the pristine in-plane mirror symmetries of the constituent layers are usually preserved at the interface. On-demand manipulation of these symmetries remains challenging. Here, we demonstrate a strategy to control the in-plane mirror symmetries of interfaces by engineering the crystallographic orientation of heterostructures. We design a workhorse system with a new orientation, i.e., the $LaAlO_3/SrTiO_3$ heterostructure with metallic interfaces in the (112)-plane. Such a high index orientation leads to the breaking of all the pristine mirror symmetries except the mirror plane perpendicular to the $[1\bar{1}0]$ direction ($M_{[1\bar{1}0]}$), resulting in the $C_s$ point symmetry with a metallic conduction. Consequently, this interface exhibits a giant nonlinear Hall effect characterized by a large Berry curvature dipole, a circular photogalvanic effect, and current-induced out-of-plane magnetization, all functional at room temperature. The magnitude of the nonlinear Hall effect rivals the Weyl and Dirac systems. Our work establishes a new strategy in exploring emerging electronic properties with nontrivial quantum geometry by designing the interface symmetry.

Mirror symmetry plays a fundamental role in physical effects and materials properties, especially the orbital hybridization and spin-orbital coupling[1]. Thus, it controls the geometric property of the material's electronic wavefunction, especially the Berry curvature and its dipole[2–5], and determines the gyrotropic behaviors characterized by second-order axial tensors, including the circular photogalvanic effect[6] and (inverse) spin-galvanic effect[7,8]. For example, the integral of the Berry curvature dipole (BCD) over the $k$-space in the two-dimensional system, where BCD represents the first moment of the Berry curvature, persists only if there is no more than one mirror symmetry[9]. Similarly, only in the same condition, an in-plane charge current at the interface generates magnetization with an out-of-plane moment via the inverse spin-galvanic effect[10,11]. The latter is critical for the field-free magnetization switching by spin-orbital torque[12]. Moreover, the same in-plane mirror symmetry breaking induces nonlinear electronic transport effects, such as the nonlinear Hall effect (NLHE)[9,13,14] and bulk photovoltaic effect[6]. Usually, semiconductive and metallic materials possess highly symmetric structures. Therefore, the above stringent symmetry requirement confines the manifestation of these intriguing effects in a very limited range of materials, mainly in several two-dimensional layered materials, such as Weyl semimetal $WTe_2$[13,14] and $TaIrTe_4$[15], corrugated bilayer graphene[16], and $WSe_2/SiP$[17]

[1]Hefei National Laboratory, Hefei, Anhui, China. [2]International Center for Quantum Design of Functional Materials (ICQD), Hefei National Research Center for Physical Sciences at the Microscale, University of Science and Technology of China, Hefei, China. [3]Department of Physics, The University of Warwick, Coventry, UK. [4]School of Emerging Technology, The University of Science and Technology of China, Hefei, China. [5]Department of Physics, University of Science and Technology of China, Hefei, China. [6]Center for Emergent Matter Science (CEMS), RIKEN, Wako, Saitama, Japan. [7]These authors contributed equally: Hang-Bo Zhang, Zhen-Yu Ding. ✉e-mail: ygao87@ustc.edu.cn; mingminyang@hfnl.cn

and WSe₂/BP[18] heterostructures. Moreover, the above physical effects in these materials generally function at low temperatures with limited exceptions[15,19]. It is therefore highly desirable to develop a strategy that can deliberately tailor the mirror symmetry in materials, especially those with metallic conduction and induce the above effects.

Here, we demonstrate exactly such a strategy based on the engineering of the crystallographic orientation of any heterostructure. Although this approach generates any desired mirror symmetry at the interface and is both handy and versatile, it has been overlooked until now. Crystallographic orientation engineering has already been the used technique in bulk materials, especially piezoelectrics[20], to achieve optimized electromechanical coupling and minimized loss. However, how the crystallographic orientation of heterostructures affects the symmetry of the interfacial region remains elusive, let alone the implications on physical properties. To unravel its hidden potential, we choose as a model system, the well-known LaAlO₃/SrTiO₃ hetero-structure, which has garnered tremendous attention since its discovery[21]. Canonical low-Miller-index LaAlO₃/SrTiO₃ interfaces with (001), (110) and (111) orientations, respectively possess $C_{4v}$, $C_{2v}$, $C_{3v}$ point symmetry, retaining at least two in-plane mirror symmetries at room temperature. As a consequence of the built-in symmetries, these interfaces can generate current-induced magnetization with only in-plane moment[22] and the circular photogalvanic effect (CPGE) only works with inclined illumination[23]. Also, this interface exhibits a

negligible nonlinear Hall effect at room temperature. As recently shown, LaAlO₃/SrTiO₃ two-dimensional electron gas (2DEG) interface shows sizable nonlinear transport properties at low temperatures due to a lower symmetry induced by a phase transition and ferroelectric instability in SrTiO₃ crystal[24]. It can be anticipated that once the unwanted pristine mirror symmetries are removed, the LaAlO₃/SrTiO₃ interface may exhibit unprecedented physical effects and functionalities even at room temperature or above. In this regard, we have demonstrated here that forming interfaces on crystal planes of high Miller index (11X) enables the rational breaking of the in-plane mirror planes and emergence of hidden properties. Taking the (112)-LaAlO₃/SrTiO₃ as a model example, this engineered interface exhibits a suite of intriguing phenomena—giant nonlinear Hall effect (featuring a substantial Berry curvature dipole), circular photogalvanic effect, and current-induced out-of-plane magnetization—all operative at room temperature and above.

## Results and Discussions

We grew a thin LaAlO₃ layer on (112)-oriented SrTiO₃ substrates using the pulsed laser deposition technique (see Fig. 1a and Methods). The thin film exhibits an ultrasmooth surface. The X-ray diffraction (XRD) analysis revealed a high crystalline quality indicated by the presence of Laue oscillation peaks (Supplementary Note 1). The conductive electron layer confined at the interface of LaAlO₃/SrTiO₃ heterostructures

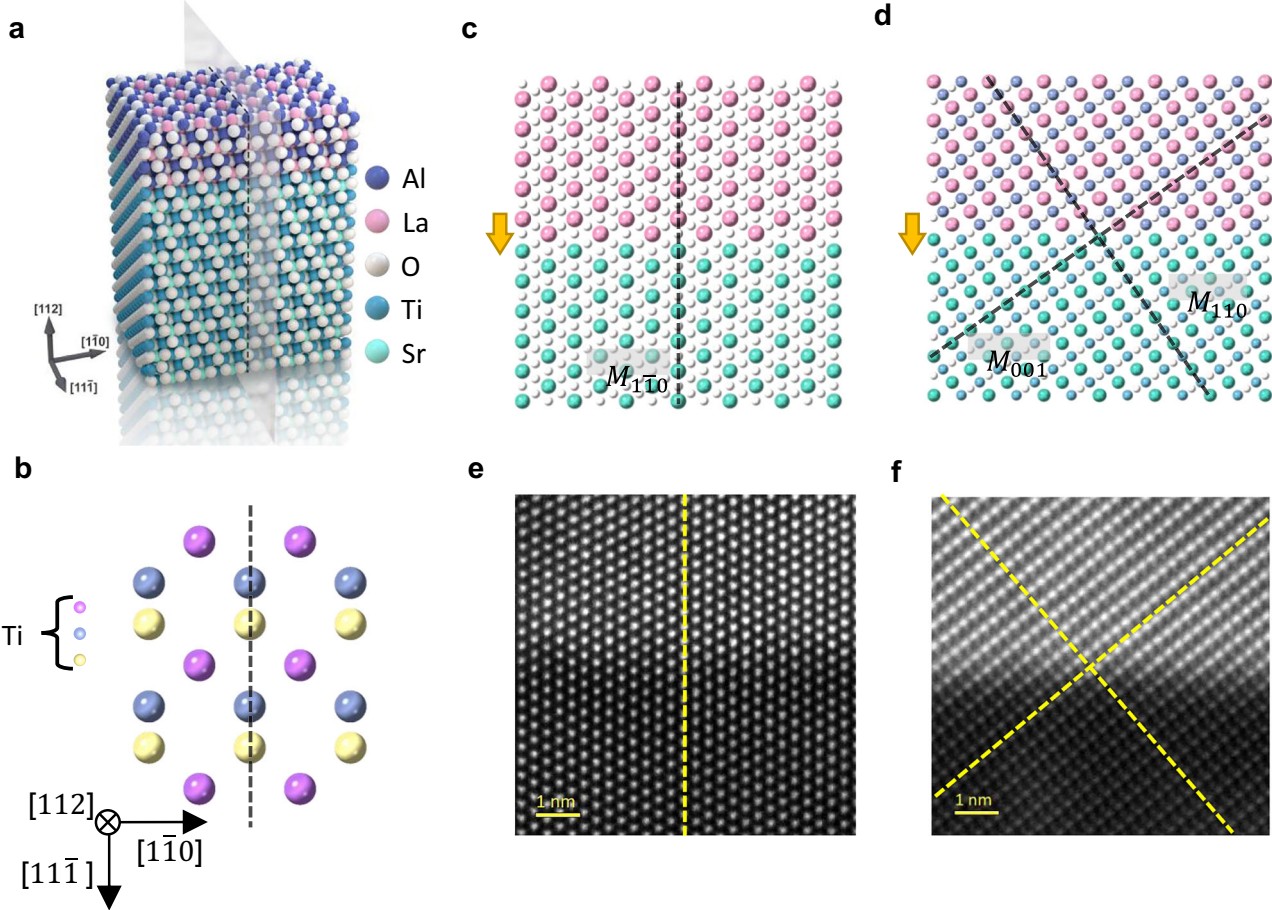

**Fig. 1 | Symmetry engineering of (112)-LaAlO₃/SrTiO₃ interface. a** An atomic sketch of the heterostructure of (112)-LaAlO₃/SrTiO₃ system. **b** Top view of the Ti atoms distribution along the [112] crystallographic direction. Only one mirror plane is preserved, as shown in the dashed line. **c, d** The two-dimensional sketch of atomic distribution along (**c**) [11$\bar{1}$] and (**d**) [1$\bar{1}$0] in-plane directions. Black dashed lines indicate the mirror planes in pristine SrTiO₃ crystals. Yellow arrows indicate the

out-of-plane polarity at the interface. Only $M_{[1\bar{1}0]}$ aligns with the out-of-plane polarity, thus preserves. **e, f** Atomic resolved ADF-STEM results show the interface between LaAlO₃ and SrTiO₃ layers along the direction (**e**) [11$\bar{1}$] and (**f**) [1$\bar{1}$0] in-plane directions. Yellow dashed lines indicate the preserved mirror symmetry $M_{[1\bar{1}0]}$ in (**e**) and the broken mirror symmetry $M_{[110]}$ and $M_{[001]}$ in (**f**).

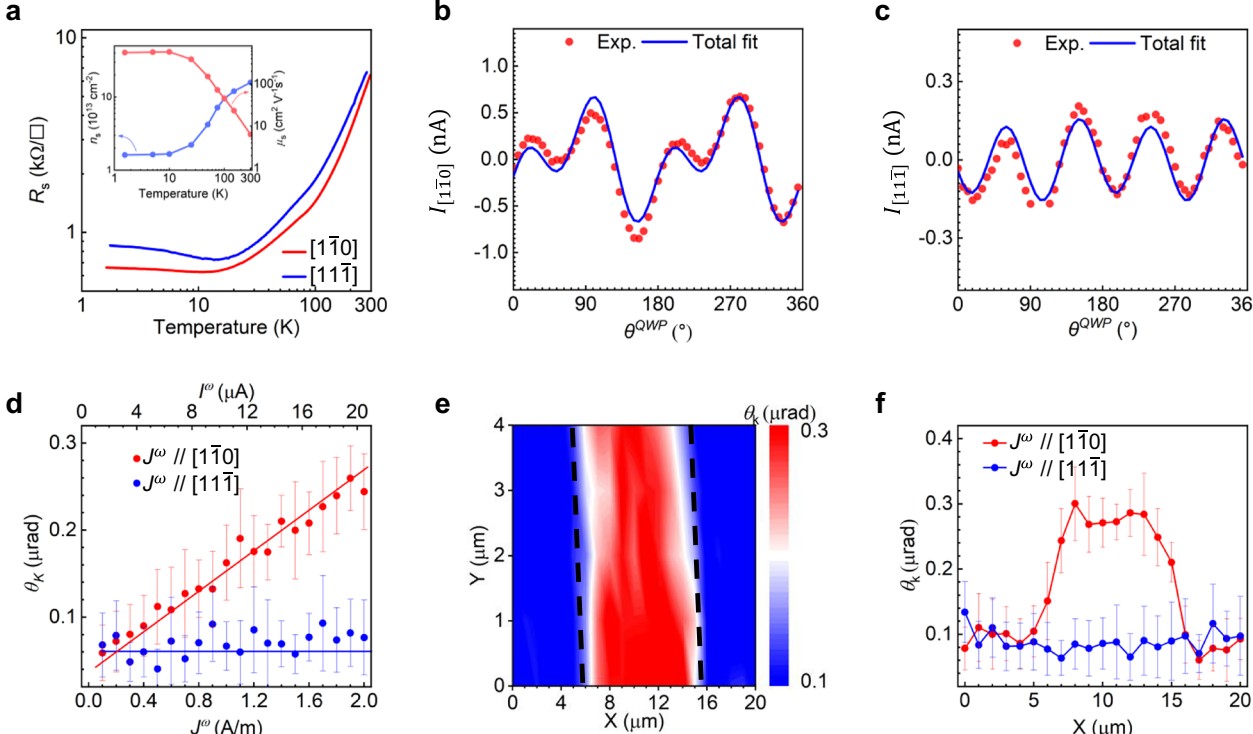

**Fig. 2 | Conductivity, CPGE, and magnetization generation at (112)-LaAlO₃/ SrTiO₃ interface. a** Sheet resistance $R_s$ as a function of temperature along two in-plane orthogonal directions. The inset shows the temperature-dependent carrier density and mobility obtained by traditional Hall measurements in the direction $[1\bar{1}0]$. **b, c** Circular photogalvanic effect studies of (112)-LaAlO₃/SrTiO₃ in the in-plane direction (**b**), $[1\bar{1}0]$ and (**c**), $[11\bar{1}]$. **d** Kerr rotation $\theta_K$ as a function of current amplitude in directions of $[1\bar{1}0]$ and $[11\bar{1}]$. **e** Mapping images of $\theta_K$ across a 10 μm-width channel at a current density of $J^\omega = 2A/m$ (20 μA) in the direction of $[1\bar{1}0]$. **f** The line scan of $\theta_K$ in the direction of $[1\bar{1}0]$ and $[11\bar{1}]$ at a current density of $J^\omega = 2A/m$. The error bar of each data point is statistically calculated over 600 measurement values.

accompanies a potential gradient along the out-of-plane direction, making the interface of polar nature[25–27] (see Supplementary Note 2 for discussion on the interfacial polarity). A distinctive feature of the (112)-orientation is that only the $M_{[1\bar{1}0]}$ mirror symmetry of SrTiO₃ crystal is preserved as it aligns with the out-of-plane direction, i.e., the interface polarity (Fig. 1a, b). In contrast, all the other intrinsic mirror symmetries of pristine SrTiO₃, such as $M_{[110]}$ and $M_{[001]}$, intersect with the out-of-plane polarity and are consequently broken at the interface region (see Fig. 1c, d). Thus, the (112)-interface possesses only one mirror plane, resulting in the $C_s$ point symmetry group (see Supplementary Note 3 for further discussion on the mirror symmetry). Note that one can further remove the residual mirror symmetry by using a different orientation (1XY) wherein $X \neq Y \neq 1$. Detailed mathematical analysis of the mirror symmetry breaking at the (11X)-interface is given in the Methods section.

The atomic arrangement at the engineered interface was characterized by scanning transmission electron microscopy (STEM). Figure 1e, f correspond to annular dark field (ADF) STEM images of the (112)-LaAlO₃/SrTiO₃ thin film along $[11\bar{1}]$ and $[1\bar{1}0]$ in-plane directions, respectively. The atomic arrangement visualized by STEM is consistent with the model shown in Fig. 1c, d. The mirror-symmetrical arrangement of atoms is clearly shown in the $(11\bar{1})$-plane corresponding to the $M_{[1\bar{1}0]}$ symmetry (Fig. 1e), whereas the tilted atomic arrangement in the $(1\bar{1}0)$ cross-section plane confirms the absence of mirror symmetry therein (see Fig. 1f).

To confirm that a metallic interface with a monoclinic $C_s$ symmetry is formed at the (112)-LaAlO₃/SrTiO₃ heterostructure, we measured the temperature dependence of its sheet resistance $R_s$ and the Hall effect. Sheet resistance $R_s$ along both $[1\bar{1}0]$ and $[11\bar{1}]$ in-plane directions show the metallic behavior with $R_s$ decreasing with temperature down to ~20 K (Fig. 2a). The small upturn of $R_s$ is likely due to

the enhanced electron scattering at lower temperatures[28,29]. Notably, $R_s$ shows different magnitudes along two in-plane directions in the temperature range investigated, which is consistent with the in-plane anisotropy of the designed symmetry. The Hall measurements (see details in Supplementary Note 4) reveal that the carrier mobility is as high as ~500 cm²V⁻¹s⁻¹ and the carrier density decreases to ~1×10¹³ cm⁻² at low temperatures, which are similar to the canonical low-Miller-index LaAlO₃/SrTiO₃ interfaces (inset of Fig. 2a)[24].

To get an insight into the mirror symmetry breaking in the designed (112)-interface, we have measured the CPGE on the same samples by illuminating the surface using a circularly polarized 405 nm laser beam with normal incidence (see Methods and Supplementary Note 5). In this particular illumination condition, the CPGE current can be generated only along the direction perpendicular to the mirror plane. Consequently, if there are more than two in-plane mirror symmetries in the sample, there will be no in-plane CPGE current generated in the heterostructure. Figure 2b shows the photocurrent flowing along the in-plane $[1\bar{1}0]$ direction (termed $I_{[1\bar{1}0]}$) as a function of $\theta^{QWP}$, where $\theta^{QWP}$ is the angle between the fast axis of the quarter-wave plate (QWP) and the polarization direction of the incident linear polarized light. We extract the CPGE contribution by fitting this curve using[17]:

$$I = C \sin\left(2(\theta^{QWP} - \theta^{off})\right) + L \sin\left(4(\theta^{QWP} - \theta^{off})\right) + I_0 \qquad (1)$$

Where $I$ is the photocurrent, $C$ and $L$ denote the CPGE and linear photogalvanic effect (see Methods) current amplitude, respectively; $\theta^{off}$ is the angle between the fast axis of the QWP and the crystallographic direction $[1\bar{1}0]$ (see details in Supplementary Note 5). $I_0$ is the background photocurrent independent of the light polarization[6]. This revealed a sizable CPGE current $C_{[1\bar{1}0]}$ along the $[1\bar{1}0]$ direction

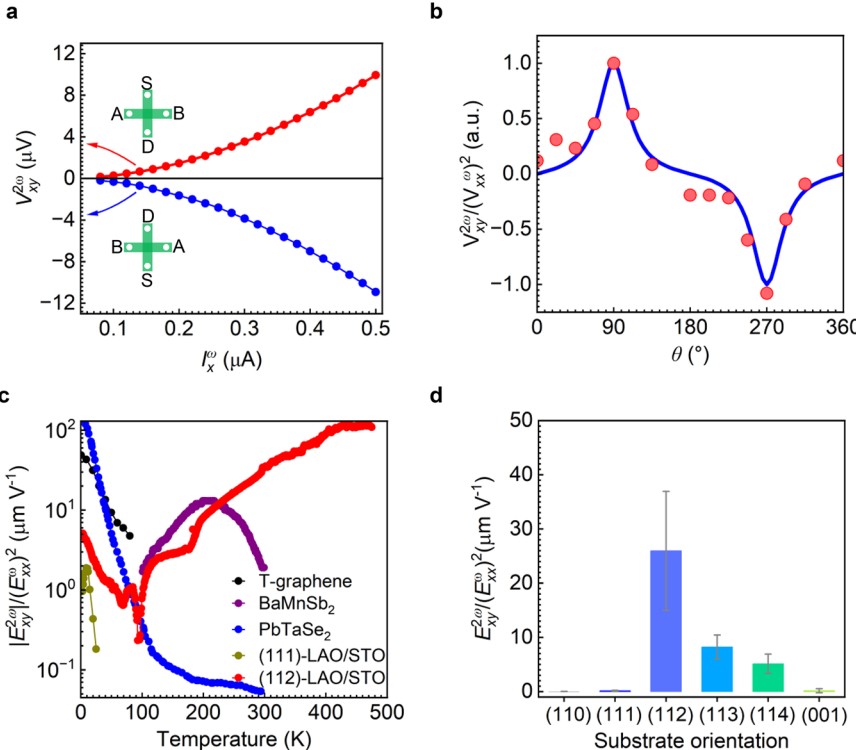

**Fig. 3 | Nonlinear Hall effect of (112)-LaAlO₃/SrTiO₃. a** Second-harmonic transverse $V_{xy}^{2\omega}$ in response to the forward (upper panel) and backward (lower panel) a.c. current along $[1\bar{1}0]$. Insets show the measurement configuration. **b** Nonlinear Hall response as a function of bias current direction $\theta$ measured from a disc device. **c** Comparison of nonlinear Hall coefficients of twisted graphene (T-graphene)[32],

BaMnSb₂[19], PbTaSe₂[33], (111)-orientated LaAlO₃/SrTiO₃ ((111)-LAO/STO)[24] and our work (112)-LaAlO₃/SrTiO₃. **d** Substrate orientation-dependent nonlinear Hall effect at room temperature. Error bars are the mean standard deviation of values obtained over at least 4 devices.

with an amplitude of ~0.4 nA (Supplementary Note 5). In contrast, there generates a negligible CPGE current (0.02 nA) along the $[11\bar{1}]$ direction (Fig. 2c), likely due to the misalignment of the electrodes. Therefore, all the above results confirm a conductive layer with $C_s$ symmetry emerging at the interface of two centrosymmetric insulators with (112)-orientation.

The interface with such a low symmetry would inevitably possess various physical effects that are normally absent in the well-studied (001), (110) and (111)-orientations. We focus on three such effects shown by the (112)-interface: the charge-to-magnetization conversion, i.e., the inverse spin-galvanic effect, the nonlinear Hall effect and the quantum geometric feature of the electronic wavefunction at the interface. These effects and properties are both of fundamental interest and technological importance. The peculiarity of the present (112)-interface is that a nonequilibrium magnetization with an out-of-plane component can be generated by a current flowing in the $[1\bar{1}0]$ in-plane direction (see Methods). Whereas the current flowing in the perpendicular $[11\bar{1}]$ direction can only generate an in-plane magnetization, as happens in the conventional (001), (110) and (111)-orientations. The current-induced out-of-plane magnetization in the (112)-interface is probed by the polar magneto-optical Kerr effect (MOKE). The experimental setup is shown in Supplementary Note 6 and discussed in Methods. The MOKE signal, i.e., Kerr rotation $\theta_K$ as a function of the a.c. current amplitude is given in Fig. 2d (see Methods). A sizable MOKE signal appears when the current runs along the $[1\bar{1}0]$ direction, of which the magnitude increases almost linearly with the current density. In contrast, only a small background noise signal is shown when the source current is driven along the $[11\bar{1}]$ direction or other orientation interfaces, which is in line with the theory (Supplementary Note 7). A spatial distribution of the out-of-plane magnetization has been probed by mapping the polar MOKE signal across the Hall bar (Fig. 2e). Clearly, the almost uniform out-of-plane magnetization is

generated over the Hall bar (also see Fig. 2f for line scan results). This is distinctive from the spin/orbital Hall effect that only generates a polar MOKE signal with the opposite sign at the edges of the bar[30]. Such a current-induced out-of-plane magnetization can be potentially used to switch magnetization by spin-orbital torque.

In addition to the magnetization that is linearly proportional to current, a nonlinear Hall voltage also develops across the Hall bar of the (112)-oriented conductive interface with a quadratic dependence on the current. To characterize this nonlinear transport, we employed two standard Hall bar devices aligned along the two main in-plane crystallographic axes, i.e., $[1\bar{1}0]$ and $[11\bar{1}]$ (Fig. S11). With an a.c. current $I_x^\omega$ injected, both first- and second harmonics of longitudinal $V_{xx}$ and transversal voltages $V_{xy}$ of the Hall bar are measured using a lock-in amplifier (see Methods). In the case of the conductive interface with $C_s$-symmetry, a nonlinear Hall voltage can be generated only along the $[11\bar{1}]$ direction driven by an a.c. current flowing along the $[1\bar{1}0]$ direction. Figure 3a shows the room temperature second harmonic Hall voltage $V_{xy}^{2\omega}$ increases quadratically with $I_x^\omega // [1\bar{1}0]$, reaching a value of 10 µV driven by $I_x^\omega$ with an amplitude of 0.5 µA. The magnitude and phase of $V_{xy}^{2\omega}$ are almost independent of the frequency of $I_x^\omega$, excluding the contribution of capacitive coupling and thermal-related effect (Fig. S11). Moreover, the sign of $V_{xy}^{2\omega}$ reverses when the bias direction and the Hall voltage probe switch simultaneously.

In addition, the electronic transport of the (112)-orientation shall be strongly anisotropic. To show this, we characterized the tensorial nature of the transport properties of the $C_s$-electron gas using a disc-style electrode with an angle interval of $22.5°$, which allows us to drive $I_x^\omega$ along various in-plane directions with an angle of $\theta$ (Fig. S12). The two-fold angle-dependent in-plane resistance agrees well with the $C_s$ symmetry (see Methods). Meanwhile, the $I_x^\omega$ direction dependence of the nonlinear Hall voltage normalized by the square of the driven voltage (i.e. $V_{xy}^{2\omega}/(V_{xx}^\omega)^2$) shows a one-fold angular $\theta$ dependence,

which is similar to that of WTe$_2$[31] (Fig. 3b). This dependence can be well fitted by the formula derived by symmetry-based phenomenological analysis (see Methods).

To compare with all other systems, the NLHE coefficient is defined as $E_{xy}^{2\omega}/(E_{xx}^{\omega})^2$ with $E_{xy}^{2\omega}$ denotes the field induced by the Hall voltage and $E_{xx}^{\omega}$ refers to the longitudinal driving electric field. For our particular case, the NLHE exhibits three important features: (i) a remarkably wide temperature range, (ii) a very high value of the NLHE coefficient, and (iii) tunability by crystal orientation. Firstly, the NLHE persists in a wide temperature range from 3 K to 475 K (Fig. 3c). With decreasing temperature from 475 K to about 100 K, the NLHE coefficient keeps decreasing and finally reaches nearly zero. This might be due to the decreased electron density in the conduction band that shifts the Fermi level from the $e'_g$ band towards the lower $a_{1g}$ band with a decreased magnitude of BCD as discussed later. Further reducing the temperature to 3 K, the NLHE coefficient bounces back to a value slightly higher than the (111)-oriented system and comparable to Bi$_2$Se$_3$ (Fig. 3c and Fig. S14). The temperature-dependent behaviour differs from twisted graphene[32], PbTaSe$_2$[33] and other 2D materials, of which the NLHE shows a monotonic dependence on the temperature (see discussion in Supplementary Notes 10 and 11). The very high temperature range approaching 500 K sets a new remarkable benchmark in both the operational temperature range and the NLHE coefficient value, outperforming all previously reported materials.

As mentioned, all the (11$X$) crystallographic oriented interfaces shall also show similar mirror symmetry breaking. Therefore, in addition to the (112)-orientation, we have explored LaAlO$_3$/SrTiO$_3$ heterostructures with (113) and (114) orientations (Fig. 3d). All these oriented heterostructures possess conductive interfaces with $C_s$ symmetry and exhibit sizable NLHE with the room temperature coefficient value decreasing from (112) to (114). In contrast, NLHE vanishes in (001)-, (110)- and (111)-oriented heterostructures at room temperature (Supplementary Note 12). It is noted that defects such as oxygen vacancies or zigzag patterns may form differently at particularly high Miller-index interfaces during growth. However, as detailed in Supplementary Note 13, the fundamental origin of both the magnetization generation and the giant nonlinear transport is the designed symmetry breaking imposed by crystallographic-orientation engineering. Defects can influence the transport behavior, but their role is derivative—arising from, and modulated by, the underlying orientation-induced symmetry engineering.

Under the relaxation-time approximation, there are three intrinsic contributions to the second-order NLHE, i.e., the Berry-curvature dipole (BCD) mechanism[9], the quantum-metric dipole mechanism[34], and the Drude mechanism[35]. Although all three mechanisms require inversion-symmetry breaking, the last two mechanisms require additional breaking of time-reversal symmetry, which is not the case for our non-magnetic LaAlO$_3$/SrTiO$_3$. Thus, only the BCD can be the intrinsic contribution to the NLHE.

Additionally, extrinsic contributions, including the side jump and skew scattering, also play an important role in the NLHE of the (112)-interface[36]. Detailed analysis of various contributions by the scaling law is given in Supplementary Note 10. It indicates that the electrons of the (112)-interface possess a substantial Berry curvature dipole in their momentum space.

To gain an insight into the quantum geometrical feature of the conductive (112)-interface, we have performed density functional theory calculations to unravel its characteristics, especially the feature of Berry curvature and BCD (see Methods). A superlattice structure was used to model the (112)-LaAlO$_3$/SrTiO$_3$ interface (see Supplementary Note 14). The space group symmetry of the entire superlattice structure is $Pm$, corresponding to the $C_s$ point group symmetry, which is consistent with that of the (112)-oriented interface under consideration. Our calculations reveal the formation of 2DEG at the interface of the (112)-LaAlO$_3$/SrTiO$_3$ superstructure. The conductive

properties of the 2DEG are predominantly contributed by the Ti-3$d$ ($d_{xy}$, $d_{xz}$ and $d_{yz}$) orbitals at the $n$-type interface (see Supplementary Note 14). The three main bands near the $\Gamma$ point in the $K_{xy}$ plane of the 3D band structure is shown in Fig. S20 and a 2D band structure with Berry curvature $\Omega_z$, is shown in Fig. 4a.

For the 2DEG at the (112)-oriented interface, the breaking of inversion symmetry ($P$) leads to a nonzero local Berry curvature ($\Omega$), and the time reversal symmetry dictates that $\Omega_z(-\boldsymbol{k}) = -\Omega_z(\boldsymbol{k})$. Furthermore, the mirror symmetry $M_y$ requires $\Omega_z(k_x, k_y) = -\Omega_z(k_x, -k_y)$. Here, the $x$, $y$, and $z$ axes correspond to the $[11\bar{1}]$, $[1\bar{1}0]$, and $[112]$ crystallographic directions, respectively. Figure 4b illustrates the $k$-space distribution of the $\Omega_z$, clearly consistent with the symmetry analysis. It is highly nonuniform, with hot-spot areas near Fermi lines. More precisely, the Berry curvature exhibits a dipolar distribution about $k_y = 0$, as allowed by the $C_s$ symmetry. Such texture can be captured by the BCD, defined as:

$$D_{\alpha z} = \int \frac{d^2\boldsymbol{k}}{(2\pi)^2} D_{\alpha z}(\boldsymbol{k}) = \int \frac{d^2\boldsymbol{k}}{(2\pi)^2} \sum_n f_n \frac{\partial \Omega_z^n(\boldsymbol{k})}{\partial k_\alpha} \qquad (2)$$

where $f_n$ denotes the Fermi distribution function for the $n$-th band. The time reversal symmetry dictates that $D(-\boldsymbol{k}) = D(\boldsymbol{k})$; the $C_s$ point group symmetry further requires that $D_{xz}(k_x, k_y) = -D_{xz}(k_x, -k_y)$ and $D_{yz}(k_x, k_y) = D_{yz}(k_x, -k_y)$. Consequently, only $D_{yz}(\boldsymbol{k})$ can integrate to a nonzero value over the Brillouin zone. In Fig. 4c, we illustrate the $k$-space distribution of the $D_{yz}$, which is consistent with the symmetry analysis, confirming the validity of our calculation. Similar to the Berry curvature, larger $D_{yz}(\boldsymbol{k})$ appears near the Fermi lines. Notably, $D_{yz}$ reaches a sizable value near the Fermi lines and exhibits symmetries with respect to both $k_x = 0$ and $k_y = 0$, resulting in a finite BCD integrated over the $k$-space. Since the BCD is a property of the Fermi surface, it substantially depends on the position of the Fermi level in the band structure, as shown in Fig. 4d. The BCD magnitude only shows large values near the Fermi level, where two maxima of opposite signs can be found. In contrast, $D_{xz}$ remains nearly zero across the entire Fermi level, consistent with mirror symmetry $M_y$. Note that the peak value of calculated BCD is at the same order of magnitude as that of Weyl semimetal and topological insulators[13,15,37]. Note that the role of spin-orbit coupling in the geometric features has been discussed in Supplementary Note 15.

In conclusion, we have demonstrated that the crystallographic orientation engineering is a feasible approach to achieve a designed interface symmetry in heterostructures, which allows the emergence of otherwise forbidden physical effects at room temperature. The (112)-oriented LaAlO$_3$/SrTiO$_3$ conductive interface with $C_s$ symmetry exhibits a giant nonlinear Hall effect driven by the large Berry curvature dipole, current-induced out-of-plane magnetization and circular photogalvanic effect, all generated by breaking the in-plane mirror symmetry. The developed strategy is also applicable to other types of heterostructures, including metal-oxide-semiconductor field-effect transistors, tricolor superlattices and thin films with embedded strain gradient or chemical component gradient. Thus, this work opens a new material design realm that not only can be used to explore a wide range of physical effects in a low-symmetric system, such as superconductivity, opto-spintronics, spintronics, magnetoelectric and ferroelectrics, etc., but also offers the prospects to bridge these research fields that were largely decoupled before.

## Methods
### Sample and device preparation
To achieve the Ti-rich surface, (112)-SrTiO$_3$ substrates were etched in buffered oxide etchant (BOE) for 11 seconds and further annealed at 1000 °C in air for 1 h. LaAlO$_3$ thin films were epitaxially grown on

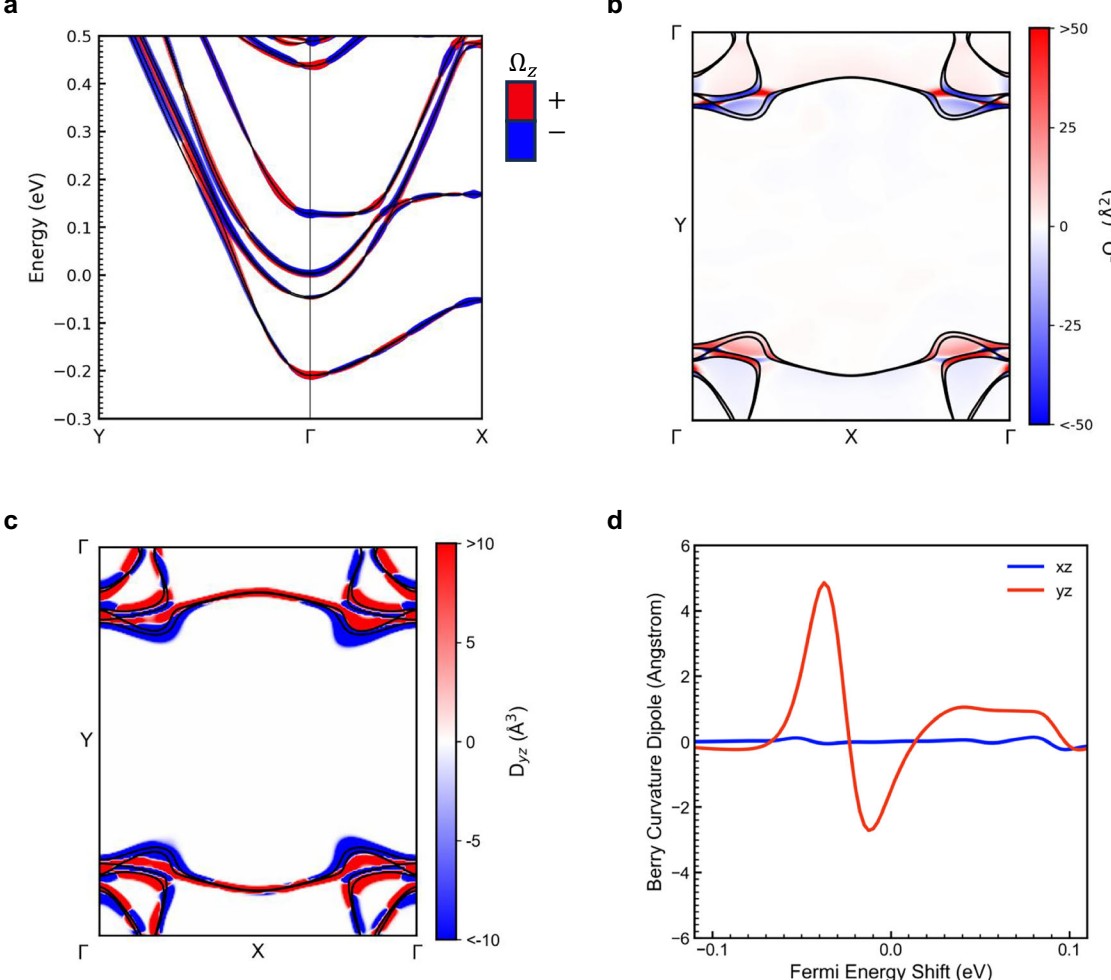

**Fig. 4 | Density functional theory calculations of the (112)-oriented LaAlO₃/SrTiO₃.** **a** 2D band structure and calculated distributions of Berry curvature $\Omega_z$, red represents positive value while blue refers to negative sign. The magnitude of the BCD is indicated by the thickness of the curves in the plot. **b**, **c** The calculated distributions of the (**b**) Berry curvature $-\Omega_z$ and (**c**) the Berry curvature dipole $D_{yz}$ in $k$-space, respectively; the black curves represent the Fermi lines. **d** The Fermi energy dependence of $D_{yz}$ and $D_{xz}$ integration in $k$-space.

SrTiO₃ substrates using pulsed laser deposition (PLD) at 800 °C in an oxygen pressure of $\sim 0.01$ Pa. The laser fluence was about $1 J/cm^2$ and the repetition rate was 2 Hz. After film growth, the samples were first cooled down to 500 °C in the same growth oxygen pressure, then subjected to 10-min in-situ annealing at ~100 Pa oxygen pressure to compensate the oxygen vacancies in the SrTiO₃ bulk. Finally, thin films were cooled down to room temperature at the same ambient oxygen pressure. We note here that different orientation LaAlO₃/SrTiO₃ thin films were grown in the same PLD condition and have a similar thickness of ~4 nm.

The photolithography technique was used to pattern Hall bar (10 μm in width and 70 μm in length) and disc devices. ~60 nm amorphous AlO$_x$ thin films grown by PLD at room temperature at an oxygen pressure of 1 Pa were used as hard masks, leaving the bare area of SrTiO₃ substrate for the active-Hall bar device growth. Disc devices were fabricated using the same method.

### Scanning transmission electron microscopy
Scanning transmission electron microscopy (STEM) specimens were prepared using Tescan Amber FIBSEM with standard lift out procedures. The cross-section specimens were then prepared along <110> and <111> directions. The specimens are polished using low accelerating voltage after thinning to remove debris and amorphized region. Atomic-resolution STEM images were acquired using a double

aberration-corrected JEOL ARM200F working at 200 kV. A probe forming a convergence semi-angle of 23 mrad was used to obtain STEM images. Annular dark field (ADF) images were formed using the signal collected from the semi-angle range 45 to 180 mrad.

### Circular photogalvanic effect measurements
A homemade optoelectronic measurement setup equipped with optical instruments microscopy cryostat was used for circular photogalvanic effect studies (Fig. S6 and S7). A polarised 405 nm laser (~37 mW nominal power) was employed to generate linearly polarised light. A quarter wave-plate (QWP) was used to vary the light polarization. The photocurrent was read by a transimpedance amplifier and digitized by a data acquisition system. It was further used to record the photovoltage as a function of the QWP angle.

### Magneto-optical Kerr effect studies
The Magneto-optical Kerr effect (MOKE) measurements (Fig. S9) were conducted in a dark environment with the temperature of optical components well controlled by a home-made system (within ± mK fluctuation). A 632 nm HeNe laser (power of ~3.5 mW) linearly polarized by a Glan-Taylor Calcite polarizer was used as an excitation source. An objective lens (×20) focused the laser onto the sample, producing a laser spot of about ~1 μm. After being reflected from the

sample, the light was collected by the objective lens and further passed through the half-wave plate and then split by a Wollaston prism into two beams, i.e., reference and signal. For a better common-mode rejection performance, we adjusted the ratio between the power of the reference and signal to -1.9:1 via the half-wave plate. An auto-balanced detector was employed to analyze the intensity difference between light paths. To further enhance the signal-to-noise ratio, we have employed an a.c. source to drive the current and analyze the auto-balanced detector signal by a lock-in amplifier. Each data point shown in Fig. 2d–f was statistically calculated from a ~ 15 min measurement duration with at least 600 data points. The Kerr rotation angle was calculated following the method reported in the reference[38]. We note here that the MOKE sensitivity probably arises from four factors: (1) the finite extinction ratio (100000:1) of the Wollaston prism placed before the balanced detector and (2) the intrinsic offset/noise and finite resolution of our lock-in amplifier. (3) The electromagnetic cross-talk between different electronic components in our home-designed MOKE system. (4) The He-Ne laser used in our system exhibits a sizable fluctuation of power and polarization.

### Electronic transport measurements

LaAlO$_3$/SrTiO$_3$ samples were mounted on the chip carrier using conductive silver paste. The Hall bars and disc bars were connected to the chip carrier with aluminum wires penetrating through the LaAlO$_3$ thin film using the ultrasonic wedge-bonding technique. An a.c. current $I^\omega$ ($\omega$ is typically 17.7 Hz in this study) was sourced along the desired crystallographic direction to perform the nonlinear transport measurement. Lock-in techniques were used to simultaneously measure the first- and second- harmonic voltage drops. A low temperature magnetic measurement system with an insert was used for the transport measurements between -2 K and 300 K; a microscopy cryostat with temperature controller was employed to measure the nonlinear transport at high temperatures up to 475 K. The temperature-dependent NLHE measured ($V_{xx}^\omega$ and $V_{xy}^{2\omega}$) continuously by heating the sample at a rate of 3 K/min with the ac current biased.

### Mathematical analysis of the symmetry evolution at the orientation-engineered interface

How the interface symmetry evolves by varying heterostructure orientation can be analyzed by matrix calculation. For simplicity, we will focus on mirror symmetry here. Regarding the SrTiO$_3$ crystal of $O_h$ point symmetry, it has three axial mirror $M_{\{100\}}$ and six diagonal mirror planes $M_{\{110\}}$ of which the matrix can be respectively given as:

$$M_{[100]} = \begin{bmatrix} -1 & 0 & 0 \\ 0 & 1 & 0 \\ 0 & 0 & 1 \end{bmatrix} M_{[010]} = \begin{bmatrix} 1 & 0 & 0 \\ 0 & -1 & 0 \\ 0 & 0 & 1 \end{bmatrix} M_{[001]} = \begin{bmatrix} 1 & 0 & 0 \\ 0 & 1 & 0 \\ 0 & 0 & -1 \end{bmatrix}$$

$$M_{[110]} = \begin{bmatrix} 0 & -1 & 0 \\ -1 & 0 & 0 \\ 0 & 0 & 1 \end{bmatrix} M_{[1\bar{1}0]} = \begin{bmatrix} 0 & 1 & 0 \\ 1 & 0 & 0 \\ 0 & 0 & 1 \end{bmatrix} M_{[011]} = \begin{bmatrix} 1 & 0 & 0 \\ 0 & 0 & -1 \\ 0 & -1 & 0 \end{bmatrix}$$

$$M_{[01\bar{1}]} = \begin{bmatrix} 1 & 0 & 0 \\ 0 & 0 & 1 \\ 0 & 1 & 0 \end{bmatrix} M_{[101]} = \begin{bmatrix} 0 & 0 & -1 \\ 0 & 1 & 0 \\ -1 & 0 & 0 \end{bmatrix} M_{[10\bar{1}]} = \begin{bmatrix} 0 & 0 & 1 \\ 0 & 1 & 0 \\ 1 & 0 & 0 \end{bmatrix}$$

The coordinate of the above matrices is in the default cubic lattice and the subscript of $M_{[xyz]}$ represents the normal direction of the

mirror plane. The potential gradient developed at the LaAlO$_3$/SrTiO$_3$ heterostructure interface can be represented by a vector, of which direction is determined by the crystallographic orientation (11$X$) with $X$ being an integer and $X \geq 2$. Thus, the interface polarity can be expressed as [11$X$]. To check whether a mirror plane persists in (11$X$)-oriented interface, one can perform the mirror reflection operation on the vector $[11X]^T$. If the $[11X]^T$ retains intact, the mirror plane is preserved in the interface; otherwise, it is broken at the interface. Based on such a simple principle, we can see that only the $M_{[1\bar{1}0]}$ plane survives, since:

$$M_{[1\bar{1}0]}[11X]^T = \begin{bmatrix} 0 & 1 & 0 \\ 1 & 0 & 0 \\ 0 & 0 & 1 \end{bmatrix} \begin{bmatrix} 1 \\ 1 \\ X \end{bmatrix} = \begin{bmatrix} 1 \\ 1 \\ X \end{bmatrix} \tag{3}$$

Whereas all the other mirror symmetries are broken. Take $M_{[110]}$ as an example:

$$M_{[110]}[11X]^T = \begin{bmatrix} 0 & -1 & 0 \\ -1 & 0 & 0 \\ 0 & 0 & 1 \end{bmatrix} \begin{bmatrix} 1 \\ 1 \\ X \end{bmatrix} = \begin{bmatrix} -1 \\ -1 \\ X \end{bmatrix} \neq [11X]^T \tag{4}$$

This principle is also applicable to the rotation symmetry. One can find that all the rotation symmetry operations in the pristine SrTiO$_3$ lattice are broken at the (11$X$)-LaAlO$_3$/SrTiO$_3$. Thus, the symmetry of studied (112)-LaAlO$_3$/SrTiO$_3$ is reduced to $C_s$. Furthermore, if one chooses an orientation as (1$XY$) with $X \neq Y \neq 1$, all the mirror reflection symmetry and rotation symmetry are broken, leading to the symmetry of $C_1$.

### Phenomenological theory of circular photogalvanic effect (CPGE)

To understand the CPGE with regard to the crystallographic directions, the classical phenomenological theory is employed. The general expression of the photogalvanic effect (PGE) current is written as[6]:

$$I_i = I_{light}(\beta_{ijk}^L e_j e_k^* + i\beta_{ik}^C e_k \times e_k^*) \tag{5}$$

where $I_i$ is the photocurrent in the $i$ direction, $I_{light}$ is the intensity of the light, $\beta_{ijk}^L$ and $\beta_{ik}^C$ are the third and second order tensors describing the linear and circular PGE effect, respectively. $e_j$ and $e_k$ is the electrical field of the propagating light, $e_k^*$ is the conjugate of $e_k$. The tensor elements of both linear photogalvanic effect (LPGE) and circular photogalvanic effect (CPGE) are nonzero only in asymmetric materials. Due to the symmetry nature of the 3$^{rd}$ rank tensor of crystals, the LPGE tensor of $C_s$ point group material can be written in the form of $3 \times 6$ tensor with 10 independent nonzero elements:

$$\beta_{ij}^L = \begin{pmatrix} \beta_{11}^L & \beta_{12}^L & \beta_{13}^L & 0 & \beta_{15}^L & 0 \\ 0 & 0 & 0 & \beta_{24}^L & 0 & \beta_{26}^L \\ \beta_{31}^L & \beta_{32}^L & \beta_{33}^L & 0 & \beta_{35}^L & 0 \end{pmatrix} \tag{6}$$

The CPGE tensor of $C_s$ point group material has 4 nonzero independent elements and is expressed as:

$$\beta_{ij}^C = \begin{pmatrix} 0 & \beta_{12}^C & 0 \\ \beta_{21}^C & 0 & \beta_{23}^C \\ 0 & \beta_{32}^C & 0 \end{pmatrix} \tag{7}$$

A mixed linear-circular polarised light was generated by the laser and QWP, and hence, the expressions for LPGE and CPGE are:

$$I^L = I_{light}\beta_{ijk}^L e_j e_k^* = I_{light}\begin{pmatrix} \beta_{11}^L & \beta_{12}^L & \beta_{13}^L & 0 & \beta_{15}^L & 0 \\ 0 & 0 & 0 & \beta_{24}^L & 0 & \beta_{26}^L \\ \beta_{31}^L & \beta_{32}^L & \beta_{33}^L & 0 & \beta_{35}^L & 0 \end{pmatrix}\begin{pmatrix} \sin^2\left(2\theta^{QWP}\right) \\ \cos^2\left(2\theta^{QWP}\right) \\ 0 \\ 0 \\ 0 \\ \frac{1}{2}\sin(4\theta^{QWP}) \end{pmatrix}$$
(8)

$$I^C = I_{light}\,i\,\beta_{ik}^C e_k \times e_k^* = I_{light}\begin{pmatrix} 0 & \beta_{12}^C & 0 \\ \beta_{21}^C & 0 & \beta_{23}^C \\ 0 & \beta_{32}^C & 0 \end{pmatrix}\begin{pmatrix} 0 \\ 0 \\ -\sin(2\theta^{QWP}) \end{pmatrix}$$
(9)

where $\theta^{QWP}$ is the QWP rotation angle ($\theta^{QWP} = 0$ is in the direction $[11\bar{1}]$). The total current comprises both linear and circular PGE currents running along $[1\bar{1}0]$ and $[11\bar{1}]$ is expressed as:

$$I_{[1\bar{1}0]} = I_{light}\left(\frac{1}{2}\beta_{26}^L \sin(4\theta^{QWP}) - \beta_{23}^C \sin(2\theta^{QWP})\right)$$
(10)

$$I_{[11\bar{1}]} = \frac{1}{2}I_{light}\left(\beta_{11}^L + \beta_{12}^L - \beta_{11}^L\cos\left(4\theta^{QWP}\right) + \beta_{12}^L\cos(4\theta^{QWP})\right)$$
(11)

Hence, CPG current flows only in the $[1\bar{1}0]$ direction, while LPG current exists in both $[1\bar{1}0]$ and $[11\bar{1}]$ directions.

## Phenomenological theory of the charge-to-magnetization conversion

The current-induced nonequilibrium magnetization can be phenomenologically described by inverse spin-galvanic effect, i.e., by the formula[11]:

$$M_i = \gamma_{ij}J_j$$
(12)

Where $M_i$ is the magnetization in the direction $i$, $\gamma_{ij}$ is the second order axial tensor sharing the same symmetry form with the CPGE tensor and $J_j$ is the electric current density in the direction $j$. When an in-plane current $(J_x, J_y, 0)$ is applied to the conductive interface, the induced magnetization is thus given as:

$$M_i = \begin{pmatrix} 0 & \gamma_{12} & 0 \\ \gamma_{21} & 0 & \gamma_{23} \\ 0 & \gamma_{32} & 0 \end{pmatrix}\begin{pmatrix} J_x \\ J_y \\ 0 \end{pmatrix} = \begin{pmatrix} \gamma_{12}J_y \\ \gamma_{21}J_x \\ \gamma_{32}J_y \end{pmatrix}$$
(13)

Hence, current along $x$ ($[11\bar{1}]$) only generates an in-plane magnetization $M_y = \gamma_{21}J_x$, while the current along $y$ ($[1\bar{1}0]$) generates an in-plane magnetization $M_x = \gamma_{12}J_y$ and an out-of-plane magnetization $M_z = \gamma_{32}J_y$. Only the out-of-plane magnetization was detected in our polar-MOKE setup.

## Phenomenological theory of nonlinear transport properties

To understand the tensorial feature of the nonlinear transport of $C_s$-2DEG, we performed the phenomenological analysis. Generally, the second-order nonlinear Hall effect current induced by the electrical field $E$ is expressed as:

$$j_\alpha^{NLHE} = \chi_{\alpha\beta\gamma}E_\beta E_\gamma$$
(14)

Where $\chi_{\alpha\beta\gamma}$ is the nonlinear susceptibility tensor, and is expressed as a $3\times6$ tensor with 10 non-zero independent elements, similar to the linear PGE tensor:

$$\chi = \begin{pmatrix} d_{11} & d_{12} & d_{13} & 0 & d_{15} & 0 \\ 0 & 0 & 0 & d_{24} & 0 & d_{26} \\ d_{31} & d_{32} & d_{33} & 0 & d_{35} & 0 \end{pmatrix}$$
(15)

When an electrical field is applied to the 2D system $E_\alpha = \left(E_x, E_y, 0\right)$, the resultant NLHE current is expressed as:

$$j_\alpha^{NLHE} = \begin{pmatrix} d_{11} & d_{12} & d_{13} & 0 & d_{15} & 0 \\ 0 & 0 & 0 & d_{24} & 0 & d_{26} \\ d_{31} & d_{32} & d_{33} & 0 & d_{35} & 0 \end{pmatrix}\begin{pmatrix} E_x^2 \\ E_y^2 \\ 0 \\ 0 \\ 0 \\ 2E_xE_y \end{pmatrix} = \begin{pmatrix} d_{11}E_x^2 + d_{12}E_y^2 \\ 2d_{26}E_xE_y \\ d_{31}E_x^2 + d_{32}E_y^2 \end{pmatrix}$$
(16)

Considering the general resistance anisotropy of $C_s$ symmetry $\rho = \begin{pmatrix} \rho_{11} & 0 & \rho_{13} \\ 0 & \rho_{22} & 0 \\ \rho_{13} & 0 & \rho_{33} \end{pmatrix}$, the NLHE electrical field then is expressed as:

$$E_\alpha^{NLHE} = \rho j_\alpha^{NLHE} = \begin{pmatrix} \rho_{11}\left(d_{11}E_x^2 + d_{12}E_y^2\right) \\ \rho_{22}\left(2d_{26}E_xE_y\right) \\ \rho_{33}\left(d_{31}E_x^2 + d_{32}E_y^2\right) \end{pmatrix}$$
(17)

Where specifically, $\rho_{11}$, $\rho_{22}$, and $\rho_{33}$ refers to the resistivity in the main axis direction $[1\bar{1}0]$, $[11\bar{1}]$ and $[112]$. In our experiments, an in-plane current is sourced, read as: $J_\alpha = J\begin{pmatrix} \sin\theta \\ \cos\theta \\ 0 \end{pmatrix}$, where $\theta$ is the angle away from the mirror line.

Because $E_\alpha = \rho J_\alpha = J\begin{pmatrix} \rho_{11}\sin\theta \\ \rho_{22}\cos\theta \\ 0 \end{pmatrix}$, the nonlinear Hall electrical field orthogonal to the source current is then expressed as:

$$E_\perp^{NLHE} = J^2\rho_{11}\sin\theta\left[(d_{12} + 2d_{26})\rho_{22}^2\sin^2\theta + d_{11}\rho_{11}^2\cos^2\theta\right]$$
(18)

Hence, we have

$$\frac{E_\perp^{NLHE}}{E_{//}^2} = \frac{\rho_{11}\sin\theta\left[(d_{12} + 2d_{26})\rho_{22}^2\sin^2\theta + d_{11}\rho_{11}^2\cos^2\theta\right]}{\left(\rho_{11}\sin^2\theta + \rho_{22}\cos^2\theta\right)^2}$$
(19)

$$\frac{V_\perp^{NLHE}}{V_{//}^2} = \frac{W\rho_{11}\sin\theta\left[(d_{12} + 2d_{26})\rho_{22}^2\sin^2\theta + d_{11}\rho_{11}^2\cos^2\theta\right]}{L^2\left(\rho_{11}\sin^2\theta + \rho_{22}\cos^2\theta\right)^2}$$
(20)

Where, $L$ and $W$ are the Hall bar length and width.

Meanwhile, we have the first-order electrical field parallel and perpendicular to the current direction are expressed as:

$$E_{//} = J(\rho_{11}\sin\theta \times \sin\theta + \rho_{22}\cos\theta \times \cos\theta) = J\left(\rho_{11}\sin^2\theta + \rho_{22}\cos^2\theta\right)$$
(21)

$$E_\perp = J(\rho_{11}\sin\theta \times \cos\theta - \rho_{22}\cos\theta \times \sin\theta) = J(\rho_{11} - \rho_{22})\sin\theta\cos\theta$$
(22)

The resistivities of two-fold angular dependence are expressed as:

$$\rho_{//} = \left(\rho_{11}\sin^2\theta + \rho_{22}\cos^2\theta\right) \qquad (23)$$

$$\rho_{\perp} = \left(\rho_{11} - \rho_{22}\right)\sin\theta\cos\theta \qquad (24)$$

### First-principles calculations

The density functional theory (DFT) modeling of $SrTiO_3$/$LaAlO_3$ interfaces has been extensively studied in the literature[39,40]. In the structural modeling process, we first performed a full structural optimization of $SrTiO_3$, which was then used as the framework for constructing the superlattice. The resulting (112)-$SrTiO_{3(5.5)}$/$LaAlO_{3(6.5)}$ superlattice structure, as shown in Fig. S20, was constructed with one interface chemically modified by substituting an Al atom with a Ti atom, thereby introducing an $n$-type chemical intermixing at the interface. The other interface was left unmodified to serve as a control. Subsequently, a full relaxation of both the lattice and ionic positions was carried out, allowing the structure to relax freely along the $z$-axis while keeping the $xy$-plane constrained.

The DFT calculations were performed using the Vienna Ab initio Simulation Package (VASP)[41], adopting the Perdew-Burke-Ernzerhof (PBE) form of the generalized gradient approximation (GGA) for the exchange-correlation functional[42]. The kinetic energy cutoff for the plane-wave basis was set to 450 eV. To account for the on-site Coulomb interactions, the GGA + U method was employed[43], with $U = 11$ eV and $J = 0.68$ eV for the f orbitals of Sr, and $U = 5$ eV and $J = 0.64$ eV for the d orbitals of Ti. The larger U value for Sr was chosen to prevent the incorrect mixing of its f orbitals near the Fermi level[44]. Brillouin zone integrations were carried out using the Monkhorst-Pack scheme with a **k**-point grid of $9 \times 13 \times 3$. To facilitate high-precision calculations, the energy convergence criterion for the electronic self-consistency was set to $10^{-7}$ eV.

To obtain a high-density sampling of the electronic structure in **k**-space, the WANNIER90 package was employed to compute the maximally localized Wannier functions (MLWFs) and perform Wannier interpolation[45,46]. The MLWFs were constructed based on the non-self-consistent field (NSCF) calculation results from VASP, utilizing 360 random initial guess orbitals. Additionally, 40 extra NSCF bands were included to facilitate the disentanglement procedure. The BC $\Omega(\mathbf{k})$ is calculated as follows:

$$\Omega_\gamma(\mathbf{k}) = \epsilon_{\alpha\beta\gamma}\Omega_{\alpha\beta}(\mathbf{k}) = \sum_n \Omega_{n,\alpha\beta}(\mathbf{k}) = -2\mathrm{Im}\sum_n\sum_{m\neq n} f_n$$
$$\frac{\langle\psi_n(\mathbf{k})|v_\alpha|\psi_m(\mathbf{k})\rangle\langle\psi_m(\mathbf{k})|v_\beta|\psi_n(\mathbf{k})\rangle}{\left(E_m(\mathbf{k}) - E_n(\mathbf{k})\right)^2} \qquad (25)$$

Where, $n$ and $m$ donate the band indices, $f_n$ represents the Fermi-Dirac distribution function of the $n$-th band, $v_\alpha(\alpha = x, y, z)$ is the velocity operator, $\psi_n(\mathbf{k})$ and $E_n(\mathbf{k})$ are the Bloch wave function and energy of the $n$-th band at wave vector **k** respectively. It should be noted that $\Omega_{n,\gamma}$ represents the Berry curvature contribution from the $n$-th band at a given **k**-point, which corresponds to the results shown in Fig. 4b. Meanwhile, $\Omega_\gamma$ denotes the integration of $\Omega_{n,\gamma}$ over all Fermi-occupied states, corresponding to the results presented in Fig. 4c.

The calculation of the BCD, $D_{\alpha\beta}$, follows

$$D_{\alpha\beta} = \int [d\mathbf{k}] D_{\alpha\beta}(\mathbf{k}) = \int [d\mathbf{k}] \sum_n \frac{\partial E_n}{\partial k_\alpha}\Omega_n^\beta\left(-\frac{\partial f_0}{\partial E}\right)_{E=E_n} \qquad (26)$$

Here, the Fermi−Dirac distribution function, $f_0$ is given by

$$f_0(E) = \frac{1}{e^{(E-\mu)/k_BT}+1} \qquad (27)$$

where $\mu$ is the chemical potential, $k_B$ is the Boltzmann constant, and $T$ is the absolute temperature. As $T$ approaches zero, $-\frac{\partial f_0}{\partial E}$ converges to $\delta(E - \mu)$, which is approximated in numerical calculations using a reasonable Gaussian smearing. The above calculation of BCD differs from the experimental data, which is further discussed in Supplementary Note 15.

## Data availability

The source data underlying all main manuscript figures are provided with this paper. All other data that support the findings of this study are available from the corresponding authors upon request. Source data are provided with this paper.

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

## Acknowledgements

We acknowledge the funding support from Hefei National Laboratory. This work is supported by the Innovation Program for Quantum Science and Technology (Grant No. 2024ZD0300104 and Grant No. 2021ZD0302801). Ana M Sanchez and Eoin Moynihan acknowledge funding from the Engineering and Physical Sciences Research Council (Grant No. EP/V028596/1). We are grateful to Hong-Tao Yuan for insightful discussions and deeply appreciate Ling-Fei Wang, Da-Zhi Hou, and Juan Jiang for their instrumental support in the early stages of this project. We also thank Hefei Kejing Matl. Tech. Co., Ltd. for providing SrTiO$_3$ crystal substrates with different orientations, and Anhui Epitaxy Technology Co., Ltd. for their technical support. Part of this work was conducted at the Center for Micro and Nanoscale Research and Fabrication at the University of Science and Technology of China (USTC). We also acknowledge the technical support from the Micro-Nano Fabrication and Quantum Device R&D Platform in Hefei National Laboratory. Yi-Ning Xie acknowledges the financial support from the China Scholarship Council - University of Warwick Joint Scholarship.

## Author contributions

M.-M.Y. conceived the idea and supervised the project. H.-B.Z. and M.-M.Y. designed the experiments, established the measurement systems, and developed the phenomenological theory. H.-B.Z. grew the samples, fabricated the devices, and conducted the photocurrent, electrical transport measurements and MOKE studies. H.-B.Z. also performed X-ray diffraction and atomic force microscopy studies with assistance from Z.-H.L. Y.-N.X. and E.M. carried out the STEM measurements under the supervision of A.S. Z.-Y.D. constructed the model and performed density functional theory calculations under the guidance of Y.G. and W.Z. H.-B.Z., M.-M.Y., M.A. and Y.I. wrote the manuscript with input from all authors.

## Competing interests

The authors declare no competing interests.
