## [Transparent Peer Review file · Nature Communications]

Magnetization generation and giant nonlinear transport at symmetry-engineered interfaces

Corresponding Author: Professor Ming-Min Yang

Version 0:

Reviewer comments:

Reviewer #1

(Remarks to the Author)

In this work, based on the crystallographic orientation engineering, the authors successfully fabricated the (112)-oriented LaAlO₃/SrTiO₃ conductive interface with the Cs symmetry. In particular, based on the electrical transport measurements, they demonstrated the coexistence of the intriguing current-induced out-of-plane magnetization, circular photogalvanic and giant nonlinear Hall effect effects. The results undeniably enrich the fundamental understanding of the nonlinear transport effect in oxides through crystallographic orientation engineering. Before making a decision on the paper, the authors should address the following major points:

- (1) Is it reliable to confirm the Cs symmetry of the (112)-LaAlO₃/SrTiO₃ heterostructure from the CPGE currents shown in Fig. 2(b) and (c)? This is due to the fact that the difference for the CPGE currents between (b) and (c) is not significant.
- (2) The spin-orbit coupling (SOC) effect should play a crucial role for the observed current-induced out-of-plane magnetization and nonlinear Hall effect. It seems to me that the study on the SOC of the (112)-LaAlO₃/SrTiO₃ heterostructure is missing. In the revision, I suggest the authors specify the magnitude of SOC from the analysis of experimental data and band structures. Further, it is interesting to compare the SOC parameters for different orientated LaAlO₃/SrTiO₃ heterostructures.
- (3) It is known that the nonlinear Hall effect can also be contributed by the Drude and quantum metric mechanisms. In this work, the authors focus on the contribution from the Berry curvature dipole. The authors should specify how to exclude the other possible contributions to the nonlinear Hall conductivity.
- (4) Since the BCD has been calculated as shown in Fig. 4d, I suggest the authors further calculate the nonlinear Hall conductivity and compare to experimental data.
- (5) From Fig. 3(c), the nonlinear Hall coefficient for the (112)-LaAlO₃/SrTiO₃ heterostructure increases monotonously with temperature (starting from 100 K). This is uncommon. It is known that the SOC effect will be weakened by the high temperature. If the nonlinear Hall effect is contributed by the BCD due to SOC, the nonlinear Hall coefficient should decrease with temperature. This should be further elaborated. In addition, why can the extrinsic contributions such as side jumping and screw scattering be excluded at high temperatures.

Minor points: In the main text, Fig. 3b should be Fig. 3a, Fig. 3c should be Fig. 3b etc.

Reviewer #2

(Remarks to the Author)

This study proposes a strategy for controlling in-plane mirror symmetries at oxide interfaces by engineering the crystallographic orientation of heterostructures. By utilizing a (112)-oriented LaAlO₃/SrTiO₃ interface, the high-index orientation breaks most mirror symmetries, resulting in Cs point symmetry and metallic conduction. This structural tuning enables the emergence of novel phenomena, such as a giant nonlinear Hall effect, circular photogalvanic effect (CPGE), and current-induced magnetization—all observable at room temperature. The nonlinear Hall response is comparable to that observed in Weyl and Dirac systems. The experimental data are clear and compelling; however, I am not entirely convinced by the authors' interpretation of the observed phenomena. I would like the authors to address the following points:

Figure 1e appears symmetric with respect to the [1-10] mirror plane. However, considering the effects of lattice mismatch and strain, this symmetry may not be preserved throughout the entire structure. Furthermore, if oxygen octahedral rotations are present, the symmetry would be broken. Additionally, a zigzag pattern is visible at the interface between the black and white

regions, which is typically indicative of atomic intermixing or facet formation. Therefore, the claim that a [1-10] mirror plane exists is not readily acceptable.

The observed transport anisotropy may arise from this zigzag pattern, which could also produce a stray magnetic field perpendicular to the substrate. This, in turn, may explain the current-induced magnetization.

There is no clear description of how electrical contacts were made through the LaAlO₃ layer, which is known to have high resistance.

The mobility should also exhibit anisotropy, but this is not discussed.

The definition of theta is unclear, so I hope the authors specify its reference direction. Additionally, the term "LPGE" (linear photogalvanic effect) is not defined. What is its physical origin in this system?

In Figure 2b, the fit is poor. At around 160 degrees, the experimental data deviate significantly from the fitted curve.

In Figure 2c, the CPGE component appears absent. The red fit line diverges considerably from the experimental data.

Figure 3f, which is referenced in line 194, appears to be missing from the manuscript.

The carrier concentration is not discussed. Can the position of the Fermi level account for the carrier density observed in the sample?

Reviewer #3

(Remarks to the Author)

In their manuscript titled "Magnetization generation and giant nonlinear transport at symmetry-engineered interfaces via crystallographic orientation", the authors H.-B. Zhang et al. present a series of experimental results in high-Miller-index LaAlO₃/SrTiO₃(hkl) heterostructures which is argued to derive from the specific nonvanishing Berry curvature and Berry curvature dipole densities hosted in the systems under investigation. The overall set of experiments and symmetry-based phenomenological models are consistent with the low C_s point-group of their heterointerfaces hosting the 2DEGs.

However, there are a number of experimental details which are missing, some desirable more detailed discussions regarding e.g., the peculiar temperature-dependence of their nonlinear Hall effect (NLHE), or with regards to the magnitude of the MOKE and the NLHE at different (11X) interfaces; and possibly a demonstration that the systems under investigation are gate-tunable 2DEGs.

Comment #1

#1.1. It would be useful/desireable to present in Fig.1, layer-resolved effective charges, in the ionic limit, of unreconstructed (11X)-LAO/STO heterostructures as in, e.g., the seminal work of N. Nakagawa et al. [Nature Materials 5, 204–209 (2006). <https://doi.org/10.1038/nmat1569>] in (001)-LAO/STO, or that of G. Herranz et al. [Scientific Reports 2, 758 (2012). <https://doi.org/10.1038/srep00758>] on (111) and (110)-LAO/STO interfaces. Given the (111)-LAO/STO interface is polar, I would assume that the (11X) interfaces are as well, but I did not "do the math".

#1.2. Problematically, it seems to me that the authors argue the existence of a polar interface - as a result of the potential gradient at the interface - and cite Ref. 30 [G. Singh-Bhalla et al., Nat. Phys. 7, 80-86 (2011).], c.f. lines 83-85.

Without getting into the whole debate of the "polar catastrophe scenario", this is near complete inversion of the roles of polar interfaces and existence of 2DEGs therein. The polar interface pre-exists the 2DEG. And in virtue of the electrostatic boundary condition (in a self-consistent Poisson-Schrödinger picture), there is no further confinement potential to host the 2DEG without additional charges: meaning one needs electron donors, besides the band-bending due to energy mismatched conduction and valence bands at LAO/STO interfaces. More appropriate References would instead include [W.-J. Son et al., Phys. Rev. B 79, 245411 (2009)] and [P. Delugas et al., Phys. Rev. Lett. 106, 166807 (2011)].

Moreover, it is my understanding that C. Cazorla & M. Stengel [Phys. Rev. B 85, 075426 (2012)] have since shown that for metal-capped – stoichiometric – LAO/STO systems, the LAO internal polar field is a decreasing function of LAO film's thickness, and depends on the electrostatic boundary conditions; or more precisely on the areal carrier density at both M/LAO and LAO/STO interfaces. Such that there is and I quote from them "no intrinsic built-in LAO electric field exists in the short-circuited Pt/LAO/STO capacitor system [...]" which "[...] in practice rules out the hypothesis of Zener breakdown, which was used in Ref. [G. Singh-Bhalla et al., Nat. Phys. 7, 80-86 (2011).]"

However if the authors want to argue instead that the pristine interfaces are apolar, and that interfacial polarity arises from the existence of an electrostatic confinement potential, then further discussions are needed.

#1.3. Line 113 "conventional cases" is not well defined and could be replaced by "canonical low-Miller-index LAO/STO interfaces" which I believe is what is implied. This, Ref. [29] is here maybe not the most relevant reference.

Comment #2

#2.1. The authors present in the inset of Fig. 2a, the temperature-dependence of the electronic mobility and carrier density in their 112-Lao/STO interface. It would be good to specify if those measurements were conducted in Hall bar devices; as well as disclose the raw data of their Hall effect measurements (in Extended data?), as this is the first ever report of transport properties at this specific interface.

#2.2. This opens the questions of whether the Hall effect is purely linear? And at which magnetic field value was the Hall coefficient estimated to calculate the sheet carrier densities?

#2.3. Have the authors attempted to perform a gate-tuning of the metallicity of the 2DEG at low T (below 10K), where the extremely large dielectric constant of the STO substrate should allow for an efficient electrostatic field-effect in back-gating geometry.

This is possibly beyond the scope of the present work, but would represent a substantial addition to the current manuscript.

#2.4. Obviously the corollary to the latter becomes how does the NLHE evolve as a function of sheet conductance (or back-gate voltage)?

Comment #3

#3.1. The short discussion (lines 107-109) related to small upturn in R_s “commonly existing in LAO/STO 2DEG systems” should be slightly moderated. In some sense, yes, it is commonly observed, but somewhat not in state-of-the-art, clean and optimally doped systems [c.f. G. Herranz et al., Phys. Rev. Lett. 98, 216803 (2007)]. The upturn is, without debate, a sign of increased scattering rate of electrons at low T. The underlying (microscopic) mechanism is much more difficult to pinpoint. This is generally difficult due to the general tendency to preferentially form either cationic (La, Al) or anionic/oxygen vacancies during the growth of the LAO layer, depending on the partial pressure of oxygen.

I would certainly not put “Kondo” and “commonly” side-by-side. Despite the speculated role of oxygen deficiency clusters and their associated magnetism in STO-based systems, this is highly debated in the community; see also the decades-old controversy of magnetism in d^0 (or $d^0+\delta$) electron systems. The authors should provide a couple more references for the upturn when the growth conditions are suboptimal, such as in the early reports by A. Brinkman et al. [Nature Materials 6, 493–496 (2007). <https://doi.org/10.1038/nmat1931>], or other tentative explanations based on Kondo-screening [J. Ruhman et al., Phys. Rev. B 90, 125123 (2014) <https://doi.org/10.1103/PhysRevB.90.125123>]

On the other hand, in the seminal work of G. Herranz et al. [Scientific Reports 2, 758 (2012). <https://doi.org/10.1038/srep00758>], on (111) and (110)-LAO/STO interfaces, the upturn was very pronounced for thick (111)- and (110)- interfaces, as opposed to nearly absent for thinner LAO layers. However, in later reports, this can be entirely mitigated by finding optimal growth conditions, and this thickness-dependent upturn is no longer seen as an intrinsic property of the interface.

This brings me to the following questions.

#3.2. What is the thickness of each of the LAO(hkl) layers studied in this work?

#3.3. In Extended Data Fig. 1, the authors present a typical X-Ray diffraction pattern for the (112) interface. How about all the other interfaces? Namely the LAO/STO(113), (114), (115), as well as the (110), (111) and (001) interfaces; all of which are discussed in relation to the NLHE displayed in Fig. 2d.

Comment #4

Regarding the magneto-optical Kerr effect (MOKE) data and the circular photogalvanic effect (CPGE) presented in Figure 2.

#4.1. I understand that, on one hand, the current-induced MOKE signal could be expected to be small. However, on the other hand, the sensitivity of (longitudinal) MOKE setups can reach down to a couple of nrad, so the signal reported by the authors is rather large in comparison. This brings me to the following: How come the Kerr rotation angle at the lowest probed current density (as well as throughout the whole [11-1] data set) is as large as 0.05 microrad (or 50 nrad)? Can the authors comment on the tentative origin of this relatively large offset value?

#4.2. The data in Fig. 2e is presented for a current density of 2 A/m, that is, given the 10 microns cross-section of the Hall bar, for a current amplitude of 20 microAmps. This is 40 times larger than any of the second harmonic I-V characteristic presented in Fig. 3a and Extended Data Fig. 4c. For consistency, and in the spirit of transparency, the authors could either display a secondary x-axis (top of panel 1d) with the absolute value of the current.

#4.3. Line 115 the authors claim “To show that indeed the interface possesses a monoclinic Cs symmetry, we have measured the CPGE [...]”. This is incorrect and should be rephrased. The existence of a CPGE cannot prove per se the Cs

symmetry of the system. The CPGE is instead a potentially symmetry-allowed effect, and its anisotropy consistent with Cs symmetry.

Comment #5

Last but not least, the nonlinear Hall effect data (Fig. 3).

The symmetry-based arguments made in favor of the existence of a Berry curvature dipole-driven NLHE, and the first-principle calculations associated to it, are not being questioned here. Rather:

#5.1. In relation to my previous point: what happens to the nonlinear I-V characteristics (Fig. 3a, Ext. Data Fig. 4) when sourcing current densities comparable to those used in Fig.2? And very importantly, do those NL I-V present a hysteretic behaviour?

#5.2. Same question goes for the longitudinal linear I-V characteristic, ideally $I(\omega)$ vs. $V_{xx}(\omega)$ (see Extended Data Fig. 4b) - and $I(\omega)$ vs. $V_{yy}(\omega)$ - in the same current density range)?

In general, how can the authors rule out heating related effects or other sources of nonlinearities?

#5.3. In Fig. 2c, how is each data point obtained? Do the authors fit a full nonlinear I-V characteristics at a given temperature, or is it being measured at constant current bias upon continuously sweeping the temperature (cooldown? or warming? And at which rate?).

#5.4. Could the authors comment further on the quite peculiar temperature-dependence of the NLHE reported in Fig. 3c? is there a tentative explanation for the strong dip at about 100K, and for a recovery of the effect's magnitude at lower temperature? Can this be understood in relation to the measured temperature-dependence of the sheet conductance and sheet carrier density? The argument regarding the Fermi level

Methods section:

Line 324: annealed in which conditions? In vacuo in-situ? in air? In pure oxygen flow furnace?

Are all the (11X), the (001), the (110) and (111) interfaces grown in the same conditions?

Miscellaneous:

In Fig. 1b: if I am not mistaken, the "x"-vector: [11-1] should point towards the bottom of the page, otherwise the arrow actually indicates the [-11-1] direction.

In Fig. 3a, is the y-axis not going from negative to positive values? Or is this deliberate?

Line 34: typo "Wely" instead of "Weyl".

Line 82: "ultrasmooth" could be replaced by "atomically (?) sharp", though the lack of spatially-resolved element specific EDXS (energy dispersive x-ray spectroscopy) may prevent to make that claim.

Line 181: "it exists", "it" should be deleted here.

Line 229 and 230: "Fermi lines" (twice) could be replaced with "Fermi level" or "Fermi energy" (and give its corresponding definition).

#Line 230: "exhibit symmetries concerning both $k_x = 0$ and $k_y = 0$ ": which symmetries? What is meant here by "concerning"? I am confused.

Line 250: "are" should be "were" or "have been" (due to the use of before at the end of the sentence).

Extended Data Fig. 7, what are the corresponding current densities?

On many occasions, articles such as "the" or "a/an" are missing, or on the contrary not necessary. I would urge the authors to take the necessary English language revisions.

In conclusion, I cannot recommend the present manuscript, as it is, for publication in Nature Communications, but I encourage the authors to address my comments and questions. Upon satisfying revisions, I am of the opinion that this work would be of great interest to the broader solid-state physics community with a keen interest on nonlinear transport, and their consequences for applications, in low-symmetry systems.

Version 1:

Reviewer comments:

Reviewer #1

(Remarks to the Author)

I have carefully studied the authors' response letter and revised manuscript, which have addressed my concerns. I would like to recommend the revised manuscript for publication.

Reviewer #2

(Remarks to the Author)

I would first like to thank the authors for their efforts in revising the manuscript and for addressing several of the previous concerns. While some points have indeed been clarified, there remain a number of important issues that are still insufficiently resolved:

1. Role of defects and interface structures: The authors argue that if experimental imperfections were to break the in-plane mirror symmetry and significantly alter the interface properties, nonlinear Hall effects should also be observable in the (001), (110), and (111) interfaces, which is not the case (as shown in Fig. 3d of the main text). However, one should keep in mind that the nature of defects and interdiffusion strongly depends on the growth orientations. In high-energy surfaces such as (112), different kinds of interface reconstructions are likely to occur compared to (001), (110), or (111). Although many effects may indeed average out, not all of them do. Since the manuscript does not discuss the difference in defect formation among these orientations, it is still possible that the observed effects originate from specific microstructural features unique to the (112) interface. Changes in oxygen octahedral rotations should also be considered. For example, in systems such as SRO, out-of-plane spin components have been predicted as a consequence of octahedral rotations. Therefore, the absence of nonlinear Hall effects or current-induced magnetization in (001), (110), and (111) interfaces does not necessarily prove that the observed effects in (112) originate solely from the preservation of the $M[1-10]$ mirror plane.

2. Interpretation of the zigzag pattern: The discussion of the zigzag pattern at the interface remains ambiguous. The authors conclude that such patterns do not generate physical effects, but no clear justification is given. Transport properties and spin are very sensitive to microscopic features. Furthermore, they cite Annadi et al. (Nat. Commun. 4, 1838 (2013)) to claim that zigzag features are also present in (110) interfaces. However, the zigzag reported in that work extends over only about one unit cell and is hardly visible in scanning transmission electron microscopy (STEM) images. In contrast, the zigzag modulation observed in the present (112) samples is much more pronounced, extending over several unit cells with a clear periodic composition modulation, as seen directly in STEM image (Fig. 1e). Indeed, when the contrast in Fig. 1e is enhanced, the zigzag pattern becomes very evident and is directly discernible in the STEM images. From this perspective, I find it inappropriate to consider the two cases equivalent. From this point of view, I disagree with the authors' statement: Both images demonstrate a sharp interface between the thin film and the substrate (line 99 on page 4). Such roughness is unlikely to average out in a Hall bar geometry and may instead contribute a finite value to the measured transport. Additionally, the anisotropy of this roughness, as seen in Fig. 1e,f, could itself be a possible origin of the observed transport anisotropy. I strongly encourage the authors to examine this point more carefully.

3. Definition of θ_{QWP} : Although the authors define θ_{QWP} as the angle between the fast axis of the QWP and the polarization of the incident light, this explanation is somewhat too brief. It would be helpful to provide a more explicit reference direction (e.g., relative to the crystal axes or the measurement geometry in the case of linearly polarized light) to avoid ambiguity.

4. Fitting of the circular photogalvanic effect (CPGE)/ linear photogalvanic effect (LPGE) Data (Figs. 2b, 2c, and R2.4): The authors state that the reviewer "may have misunderstood" and that the fits are clearly good. However, I did not misunderstand; my point concerned the significant discrepancies between the experimental data and the total fit, especially near 90 degrees and 150 degrees in Fig. R2.4. In fact, in Wang et al., Phys. Rev. Lett. 128, 187401 (2022), the CPGE in (111) LAO/STO interfaces is fitted with excellent agreement across all angles and intensities. The substantial mismatch observed here suggests that additional factors may be at play. Moreover, in the new version of the manuscript, the separation into LPGE and CPGE components has been removed, which makes the text difficult to follow, and Figs. 2b and 2c appear to be interchanged. These figures need to be carefully re-presented.

I am also concerned that the sum of the red (CPGE) and blue (LPGE) curves does not reproduce the black "total fit" curve in Fig. 3c (original version). For example, at around 135 degrees, the values of LPGE and the total fit differ, despite CPGE being nearly zero at this point. Is the fitting procedure correct? Has any angular offset been corrected for? If so, this should be clearly described in the manuscript.

Reviewer #3

(Remarks to the Author)

I carefully read the rebuttal of the authors to all three reviewers. I have an overall sense that the authors carefully addressed most of the concerns and took the necessary actions in revising the manuscript and in consequently expanding the content

of the supplementary information.

Thus, I recommend the publication of the present manuscript "Magnetization generation and giant nonlinear transport at symmetry-engineered interfaces via crystallographic orientation" by H.-B Zhang et al., in Nature communications. I believe it will be of great interest to many scientific communities working on nonlinear transport, and their theories, in various materials systems.

Regarding:

Action 3.2.3 & 3.2.4: "We believe current manuscript is already a complete piece of work and we would like to study the gate tunability of this system comprehensively in a separate work. We prefer to publish Fig. R3.4 separately."

My comment: I appreciate the authors' effort and openness to disclose unpublished results, and agree that this is a substantial addition which warrants a detailed investigation for a follow-up publication.

Regarding:

"To further check the linear dependence in these ordinary Hall effects, we conducted the Hall measurement with the magnetic field applied as large as ± 8 T. As demonstrated in Fig. R3.3b, the linearity of these curves at all temperature ranges is maintained up to ± 8 T. Therefore, the magnetic field of ± 3 T is sufficient for the Hall measurement to investigate the carrier density and mobility."

My comment: I respectfully disagree with the statement that the ordinary Hall effect is purely linear in the whole magnetic field range. It appears to me that the Hall effect is in fact slightly non-linear in field at the lowest temperatures. Overall I do not think this affects the conclusions of the manuscript, because this non-linearity is still quite small. However the authors do not ignore that when multiband conduction and Hall effect is involved, the low-field and high-field regions may yield very different Hall coefficients, and that their relation to the carrier density of either or both subbands is not the same. In particular, only the high-field Hall coefficient can be taken as inversely proportional to the total carrier density in the system. I would not insist either on a two-bands fit of the Hall effect alone because it is known to be problematic due to the large number of free-parameters in the model. I'm looking forward to a more detailed analysis in future studies of the Hall effect in these systems.

Version 2:

Reviewer comments:

Reviewer #2

(Remarks to the Author)

I would like to thank the authors for their efforts in revising the manuscript in response to my comments. Also, I apologize for the wording mistake in my previous Question 2.1, where I wrote 'breaking' instead of 'preservation' of the mirror planes. The additional experiments and discussions, particularly regarding defects and octahedral rotations, convincingly support the symmetry-based interpretation. I am satisfied that my concerns have been fully addressed, and I recommend the manuscript for publication.

Response Letter Version 1

We are very grateful to the Reviewers for their valuable and constructive comments. We have accordingly revised the manuscript and the supporting information. Also, we have changed the formality of this manuscript to suit for Nature communications. We believe we have positively addressed all comments made by the reviewers. A point-by-point response to the reviewers' comments and changes made in the revised manuscript are given below.

***Question 1.1:** In this work, based on the crystallographic orientation engineering, the authors successfully fabricated the (112)-oriented LaAlO₃/SrTiO₃ conductive interface with the Cs symmetry. In particular, based on the electrical transport measurements, they demonstrated the coexistence of the intriguing current-induced out-of-plane magnetization, circular photogalvanic and giant nonlinear Hall effect effects. The results undeniably enrich the fundamental understanding of the nonlinear transport effect in oxides through crystallographic orientation engineering. Before making a decision on the paper, the authors should address the following major points:*

Is it reliable to confirm the Cs symmetry of the (112)-LaAlO₃/SrTiO₃ heterostructure from the CPGE currents shown in Fig. 2(b) and (c)? This is due to the fact that the difference for the CPGE currents between (b) and (c) is not significant.

Answer 1.1: We appreciate the Reviewer's recognition of the innovation and significance of our work. We have addressed all the issues raised by the Reviewer in the response letter to make the message in this manuscript clearer.

Regarding the circular photogalvanic effect (CPGE), it is indeed a reliable method to detect the symmetry of materials of interest, especially the existence or breaking of mirror symmetry. The CPGE only functions in gyrotropic materials and is characterized by second order axial tensors. In particular, when light is shining along the interface/surface normal direction, only the current generated by the CPGE could flow in the direction normal to the mirror symmetrical plane. In other words, there would be no CPGE current if there is more than one mirror plane in the interface/surface. Due to such peculiarity, the CPGE effect has already been used as a probe to detect the symmetry as well as quantum geometric features in 2D layered materials and topological materials [ref.: Xu et al., Nat. Phys. **14**, 900 (2018); Ma et al., Nature **565**,

337 (2019); Duan *et al.*, *Nat. Nanotechnol.* **18**, 867 (2023)].

In our work, the CPGE generates a photocurrent of 390 pA in the [011] direction (marked as red curves in Fig. 2b and 2c), which confirms the breaking of the mirror plane M_{011} . In contrast, the CPGE current along the [111] in-plane direction is just about 20 pA, which is significantly (~20 times) smaller than that along the [011]. Such a small CPGE current along the [111] direction is likely due to reasons like electrode misalignment, miscut during the substrate polishing process, etc. These technical issues slightly deviate either the current flowing direction from [111] direction or the interface from the perfect (112) orientation. Therefore, the manifestation of CPGE current and its substantial in-plane anisotropy is consistent with the C_s symmetry at the (112)-oriented LAO/STO interface, which is further consolidated with the behaviours of the inverse spin-galvanic effect and nonlinear Hall effect observed in this system.

Action 1.1: We have added '20 pA' of the CPGE in the direction $[11\bar{1}]$ in the manuscript for highlighting the substantial contrast. Also, in order to make the figure clearer, we have removed the LPGE and CPGE fitting curves in Fig. 2b and 2c; instead, we have put relevant description in the main text.

Question 1.2: The spin-orbit coupling (SOC) effect should play a crucial role for the observed current-induced out-of-plane magnetization and nonlinear Hall effect. It seems to me that the study on the SOC of the (112)-LaAlO₃/SrTiO₃ heterostructure is missing. In the revision, I suggest the authors specify the magnitude of SOC from the analysis of experimental data and band structures. Further, it is interesting to compare the SOC parameters for different orientated LaAlO₃/SrTiO₃ heterostructures.

Answer 1.2: The spin-orbit coupling (SOC) effect is important to the anomalous Hall effect as reported in the literature [Ohuchi *et al.*, *Nat Commun* **9**, 213 (2018); Dugaev *et al.*, *Phys. Rev. B* **71**, 224423 (2005)]. However, in the nonlinear Hall effect that induced by BCD, the SOC effect does not play the critical role, such as demonstrated in the twisted Bilayer Graphene [Duan *et al.*, *Phys. Rev. Lett.* **129**, 18 (2022); Huang *et al.*, *Phys. Rev. Lett.* **131**, 066301 (2023)] and WTe₂ [Ma *et al.*, *Nature* **565**, 337 (2019)]. In our LaAlO₃/SrTiO₃ interface, we claim here that the SOC effect and the associated spin contribution do not play a critical role. In 2023, M. T. Mercaldo *et al.* reported that orbital degrees of freedom combined with crystal field effects can

intrinsically generate strong Berry curvature (BC) features in the 2D electron gas system, such as the LaAlO₃/SrTiO₃ interface, in the absence of SOC [Mercaldo *et al.*, *npj Quantum Mater.* **8**, 12 (2023)]. Later, E. Lesne *et al.* experimentally studied the nonlinear transport and BC features of (111)-oriented LaAlO₃/SrTiO₃ interface at low temperature [Lesne *et al.*, *Nat. Mater.* **22**, 576 (2023)]. Their results show the orbital-sourced Berry curvature dipole is two orders of magnitude larger than the spin-sourced one, indicating a minor role of the SOC effect in the nonlinear Hall effect. Regarding the charge-to-magnetization conversion, Johansson *et al.* showed theoretically that the orbital Rashba conversion efficiency should dominate in this system and be at least an order of magnitude larger than the spin effect [Johansson *et al.*, *Phys. Rev. Res.* **3**, 013275 (2021)]. Experimentally, A. E. Hamdi *et al.* showed that the orbital contribution dominates in charge-magnetization conversion process of the (001)-LaAlO₃/SrTiO₃ interface, which is about 16 times the spin effect [Hamdi *et al.*, *Nat. Phys.* **19**, 1855 (2023)]. Overall, previous research on the LaAlO₃/SrTiO₃ interface supports the dominant contribution of orbital degree of freedom and a minor role of the SOC effect.

To evaluate the role of SOC in our system, we first analysed the band structures of the (112)-LaAlO₃/SrTiO₃ heterostructure, focusing on the electronic states near the Fermi level that are very related to transport properties and BC effects. For the Berry curvature dipole (BCD) under investigation in this work, its behaviour is critically dependent on the electronic structure information within a very narrow energy range near the Fermi level, which is clearly demonstrated in the formula

$$D_{\alpha\beta} = \int [d\mathbf{k}] D_{\alpha\beta}(\mathbf{k}) = \int [d\mathbf{k}] \sum_n \frac{\partial E_n}{\partial k_\alpha} \Omega_n^\beta \delta(E_n - \mu). \quad (1.1)$$

As shown in Fig. R1.1(a), we have shown the energy bands within the range of ± 0.2 eV around the Fermi level along the $\Gamma - X$ path. The band splitting depicted here is caused by SOC. It can be observed that the band splitting is less than 10 meV across the entire plotting range, contributing the maximum band splitting (~ 10 meV) at an energy 0.13 eV above the Fermi level. In Fig. R1.1(b), we have magnified the region where the bands cross the Fermi level, revealing a band splitting of approximately 2 meV. These results indicate that the contribution of SOC to the nonlinear Hall transport behaviour in (112)-LaAlO₃/SrTiO₃ is weak, especially at room temperature.

Fig. R1.1 SOC-induced band splitting near the Fermi level in the (112)-LaAlO₃/SrTiO₃ heterostructure. (a) band structure along the Γ -X direction near the Fermi level. The red dashed box highlights the region of interest. (b) Magnified view of the boxed region in (a), showing a spin-orbit-induced band splitting of approximately 2.0 meV near the Fermi energy, as indicated by the red arrow.

To elucidate the contribution of the SOC effect on the geometrical feature of the (112)-interface, we have calculated the BC and BCD features without the SOC effect and compared them to those with the SOC effect (as shown in Fig. R1.2). Without the SOC effect, the BC and BCD shows spin degenerated features. The integrated BCD without the SOC effect shows a similar magnitude to that taking SOC effect into consideration. On the other hand, the SOC effect lifts the spin degeneracy and induces anti-crossing features in the band structure, which modifies the dependence of the BCD on the Fermi level position. In a word, our calculation indicates the major contribution of the BCD in the (112)-LaAlO₃/SrTiO₃ interface arises from the orbital degree of freedom whereas the SOC effect, which activates the spin-contribution, modifies the quantum geometrical feature in a minor way. This is consistent with previous works.

Fig. R1.2 Effect of SOC on Berry curvature and BCD distributions. Berry curvature distribution (a)

with SOC and (c) without SOC. BCD distribution (b) with SOC and (d) without SOC.

In addition, to address the reviewer's suggestion regarding orientation dependence, we performed a supplementary calculation on the (111)-LaAlO₃/SrTiO₃ heterostructure. The SOC-induced band splitting near the Fermi level reaches approximately 4.2 meV, while slightly larger than that the (112) case (see Fig. R1.3a, b).

Fig. R1.3 SOC-induced band splitting near the Fermi level in the (111)-LaAlO₃/SrTiO₃ heterostructure. (a) band structure along the Γ -X direction near the Fermi level. The red dashed box highlights the region of interest. (b) Magnified view of the boxed region in (a), showing a spin-orbit-induced band splitting of approximately 4.2 meV near the Fermi energy, larger than that of (112)-LaAlO₃/SrTiO₃.

Action 1.2: We have now put the calculation and discussion related to the SOC effects on the BCD and Berry curvature in SI.

Question 1.3: *It is known that the nonlinear Hall effect can also be contributed by the Drude and quantum metric mechanisms. In this work, the authors focus on the contribution from the Berry curvature dipole. The authors should specify how to exclude the other possible contributions to the nonlinear Hall conductivity.*

Answer 1.3: Under the relaxation-time approximation, there are three contributions to the second-order nonlinear Hall effect, i.e., the Berry-curvature dipole mechanism [Sodemann *et al.* *Phys. Rev. Lett.* **115**, 216806 (2015)], the quantum-metric dipole mechanism [Gao *et al.*, *Phys. Rev. Lett.* **112**, 166601 (2014)], and the Drude mechanism [Wang *et al.*, *Phys. Rev. Lett.* **127**, 277201(2021)]. Although all three mechanisms require inversion-symmetry breaking, the last two mechanisms require additional breaking of time-reversal symmetry [Wang *et al.*, *Phys. Rev. Lett.* **127**, 277201(2021)].

Since our system is nonmagnetic and its time reversal symmetry persists, only the Berry-curvature dipole mechanism can contribute to the nonlinear Hall effect.

Beyond the relaxation time approximation, there are also side jumping and screw scattering contributions to the nonlinear Hall effect. These extrinsic contributions will be discussed in Question 1.4.

Action 1.3: We have now put a statement in the main text marked in red saying that the quantum metric mechanism Drude mechanism only functions in systems without time reversal symmetry, which are not applicable in our LaAlO₃/SrTiO₃ interface.

Question 1.4: (1) Since the BCD has been calculated as shown in Fig. 4d, I suggest the authors further calculate the nonlinear Hall conductivity and compare to experimental data. (2) From Fig. 3(c), the nonlinear Hall coefficient for the (112)-LaAlO₃/SrTiO₃ heterostructure increases monotonously with temperature (starting from 100 K). This is uncommon. It is known that the SOC effect will be weakened by the high temperature. If the nonlinear Hall effect is contributed by the BCD due to SOC, the nonlinear Hall coefficient should decrease with temperature. This should be further elaborated. In addition, why can the extrinsic contributions such as side jumping and screw scattering be excluded at high temperatures.

Answer 1.4: Since these comments all relate to the BCD and various contributions to the nonlinear Hall effect, we would like to systematically discuss them here.

In addition to the intrinsic contribution of the BCD, extrinsic contributions, including side jumping and screw scattering, play an important and even dominant role in the nonlinear Hall effect (NLHE). To resolve various contributions, we employ the scaling method at both low (< 30 K) and high temperature (> 100 K) ranges (Fig. R1.4). Data between 30K and 100K are challenging to analyze due to the occurrence of phase transition with this range [Lesne et al., Nat. Mater. **22**, 576 (2023)]. The two curves in Fig. R1.4 are obtained by corresponding the NLHE coefficient $E_{xy}^{2\omega}/(E_{xx}^{\omega})^2$ and conductivity σ_s at each temperature. According to the theory [Du et al., Nat. Commun. **10** 3047 (2019)], the scaling law for NLHE can be written as: $E_{xy}^{2\omega}/(E_{xx}^{\omega})^2 = A_1 (\frac{\sigma_s}{\sigma_0})^2 + A_2 \frac{\sigma_s}{\sigma_0} + A_3$, where A_1, A_2, A_3 are the scaling parameters, σ_0 is the residual conductivity that we estimate its value as that measured at 1.55 K. The fitting results of

A_1, A_2, A_3 are shown in Table R1.1. Based on the NLHE scaling law, the scaling parameters are expressed as:

$$A_1 = C^{sk,2} + (C_{00}^{sk,1} + C_{11}^{sk,1} - C_{01}^{sk,1}) \quad (1.2)$$

$$A_2 = C_{01}^{sk,1} - 2C_{11}^{sk,1} + C_0^{sj} - C_1^{sj} \quad (1.3)$$

$$A_3 = C_{in} + C_1^{sj} + C_{11}^{sk,1} \quad (1.4)$$

where C_{in} is the intrinsic contribution, C_i^{sj} the side-jump, $C_{ij}^{sk,1}$ the Gaussian skew-scattering, $C^{sk,2}$ the non-Gaussian skew-scattering, and $i, j = 0 (1)$ denotes the defects (phonon) scattering source. There are seven parameters, i.e., seven different contributions in the above three equations, which is impossible to resolve quantitatively. To get a tentative insight into these contributions, here we need to take a bold but reasonable assumption.

Fig. R1.4 Scaling law analysis of the NLHE at temperature (a) lower than 30 K and (b) higher than 100 K.

Table. R1.1 Scaling parameters

	A_1	A_2	A_3
T < 30 K	7.56	-42.4	56.9
T > 100 K	90.5	-112.4	36.6

Since $C^{sk,2}$ only exist in the A_1 parameter, the non-Gaussian skew-scattering only plays a secondary role and we neglect it [J. Duan et al., *PRL* 129, 186801 (2022)]. In the temperature range below 30 K, the parameter A_1 is almost one order of magnitude smaller than A_2 and A_3 . This indicates that the screw scattering contributions play a minor role in the low-temperature region. Also, the phonon density is largely

suppressed at low temperatures; it would be reasonable to assume C_0^{sj} (i.e., side jump due to defects) is much larger than C_1^{sj} (i.e., side jump due to phonons). Thus, we take $A_2 = -42.4 \approx C_0^{sj}$. Similarly, $A_3 = 56.9 \approx C_{in}$.

In the temperature range above 100 K, the parameter A_1 shares a similar magnitude with A_2 but with opposite sign, while A_3 is about three times smaller. Based on equations (1.2) and (1.3), the opposite sign between A_1 and A_2 is most likely due to the opposite sign of $C_{11}^{sk,1}$ and $C_{01}^{sk,1}$ in these two equations. Since $C_{00}^{sk,1}$ (screw scattering due to defects) only exist in A_1 , it may play a minor role here. Moreover, since the side jump process is insensitive to the temperature, we assume C_0^{sj} remains as a constant over the whole temperature range. Thus, equations (1.2-1.4) can be approximated as:

$$A_1 = C_{11}^{sk,1} - C_{01}^{sk,1} = 90.5 \quad (1.5)$$

$$A_2 = C_{01}^{sk,1} - 2C_{11}^{sk,1} - C_1^{sj} = -70 \quad (1.6)$$

$$A_3 = C_{in} + C_1^{sj} + C_{11}^{sk,1} = 36.6 \quad (1.7)$$

Thus, we get $C_{in} = 57$, which almost equal to that derived at low temperature range, supporting above approximation.

Based on above analysis, we can reach here a qualitative understanding of the origins and temperature dependence of the NLHE in the (112)-LaAlO₃/SrTiO₃ interface. Clear, both the intrinsic and extrinsic contributions play important but compensating roles in this system. For example, the extrinsic contribution $C_1^{sj} + C_{11}^{sk,1}$ ($= -20.4$) is opposite to that induced by Berry curvature dipole ($C_{in} = 57$). The peculiar temperature dependence can be ascribed to the role of the parameter A_2 , i.e., the contribution of screw scattering and side jump. This also means that the NLHE effect of the (112)-interface is highly related to its conductivity with an approximated quartic dependence. When the temperature decreases from 300 K to 100 K, the conductivity and mobility increases, the screw scattering process is enhanced, inducing a negative contribution that compensates the contribution of the intrinsic one. At temperatures below 30 K, the density of phonons is significantly reduced and thus, the related screw scattering contributions ($C_{01}^{sk,1}$ and $C_{11}^{sk,1}$) are suppressed. In this circumstance, the side jump and intrinsic effect dominate. With further decreasing temperature towards 1.55 K, the conductivity decreases, the compensating contribution of side jump by defects is

reduced, leading to an enhanced NLHE.

We can also infer the magnitude of the Berry curvature dipole based on the value of $C_{in} = 57$ at low temperature. To this end, we employ the formula $D_{yz} = \frac{V_{xy}^{2\omega} \sigma_{xx}^3 W 2\hbar^2}{(I_{xx}^\omega)^2 e^3 \tau} = C_{in} \sigma_{xx} \frac{2\hbar^2}{e^3 \tau}$, where τ is the relaxation time, \hbar is the Dirac constant I_{xx}^ω is the sourced current, $V_{xy}^{2\omega}$ is the nonlinear Hall voltage, σ_{xx} is the (sheet) longitudinal conductivity, W is the Hall bar width. We estimate the τ by calculating $\tau = \frac{\mu_H m^*}{e}$, where m^* effective mass, μ_H is the Hall mobility. For convenience, we adopt m^* of (111)-LaAlO₃/SrTiO₃ reported, that is $3.49 m_e$, then we have τ at 1.55 K about 1.0 ps. Using the NLHE coefficient obtained, we estimated BCD to be ~440 nm at 1.55 K. The BCD magnitude derived from experimental results is about one order of magnitude larger than that of the (111)-LaAlO₃/SrTiO₃ interface at low temperature (< 30 K).

There is a gap between the BCD magnitude derived from experimental data and that from the DFT calculation. In fact, it is common that the nonlinear transport coefficient from DFT calculations are smaller than experimental result (See Refs [*E. Lesne et al., Nat. Mater.* **22**, 576 (2023); *M. T. Mercaldo et al., npj Quantum Mater.* **8**, 12 (2023); *H. Wang et al., npj Comput. Mater.* **5**, 119 (2019); *Q. Ma et al., Nature* **565**, 337 (2019)]).

In our case, to perform DFT calculations for the (112) high-index interface, we constructed a superlattice structure containing 60 atoms, that are much larger than the typical unit cells used in common studies, which usually include only a few or a dozen atoms [*J.-S. You et al., Phys. Rev. B* **98**, 121109(R) (2018); *H. Wang et al., npj Comput. Mater.* **5**, 119 (2019); *E. Wang et al., Phys. Rev. Lett.* **132**, 266802 (2024)]. While the structure is tractable for conventional DFT tasks such as structural relaxation and band analysis, the BCD calculation is considerably more challenging, as it requires extremely dense k-space sampling to achieve convergence due to the sensitivity to fine features near the Fermi level. To overcome this difficulty, we employed Wannier interpolation and carried out BCD results based on a 1000×1000 k-point mesh. We note that constructing well-localized Wannier functions for such a large supercell is itself a nontrivial task, requiring careful disentanglement and projection procedures.

Moreover, the 2D electron gas in the interface partially comes from the 3d orbitals in SrTiO₃, for which the correlation effect can be large. This correlation effect cannot be

well captured by the DFT calculation. Modelling the band scheme in theory is further complicated by ill-defined oxygen stoichiometries and high dielectric permittivity [Gabel *et al.*, *Adv. Electron. Mater.* 8, 2101006 (2022)]. Nevertheless, our DFT calculation still represents a state-of-the-art theoretical calculation of this newly developed system, offering valuable insight into its band structure, distribution of geometrical features in k-space and Fermi energy dependence of the BCD. We hope future work would bridge the gap between the DFT and experimental characterization.

Action 1.4: The above discussion of the BCD estimation based on experimental data has been added to the discussion section of the manuscript. The scaling analysis related to the temperature dependence is added to the Supplementary Information.

Question 1.5 Minor points: In the main text, Fig. 3b should be Fig. 3a, Fig. 3c should be Fig. 3b etc.

Answer 1.5: We thank the reviewer for pointing out these errors.

Action 1.5: We have revised these minor errors in the manuscript.

Response to Reviewer #2:

This study proposes a strategy for controlling in-plane mirror symmetries at oxide interfaces by engineering the crystallographic orientation of heterostructures. By utilizing a (112)-oriented LaAlO₃/SrTiO₃ interface, the high-index orientation breaks most mirror symmetries, resulting in C_s point symmetry and metallic conduction. This structural tuning enables the emergence of novel phenomena, such as a giant nonlinear Hall effect, circular photogalvanic effect (CPGE), and current-induced magnetization—all observable at room temperature. The nonlinear Hall response is comparable to that observed in Weyl and Dirac systems. The experimental data are clear and compelling; however, I am not entirely convinced by the authors' interpretation of the observed phenomena. I would like the authors to address the following points:

Question 2.1: *Figure 1e appears symmetric with respect to the [1-10] mirror plane. However, considering the effects of lattice mismatch and strain, this symmetry may not be preserved throughout the entire structure. Furthermore, if oxygen octahedral rotations are present, the symmetry would be broken. Additionally, a zigzag pattern is visible at the interface between the black and white regions, which is typically indicative of atomic intermixing or facet formation. Therefore, the claim that a [1-10] mirror plane exists is not readily acceptable. The observed transport anisotropy may arise from this zigzag pattern, which could also produce a stray magnetic field perpendicular to the substrate. This, in turn, may explain the current-induced magnetization.*

Answer 2.1: We would like to thank the Reviewer for bringing the interface symmetry into further discussion. We would like to respond to this comment from the following four aspects.

Firstly, we would like to stress that the peculiarity of the method developed in our work, i.e., engineering the crystallographic orientation of interfaces, is to *break* pristine mirror symmetry of the component lattices, represented by the SrTiO₃ crystal in our work. The main advantage of choosing (112)-orientation as an example in our work is to deliberately retain just one mirror symmetry $M_{[1\bar{1}0]}$ at the interface. By doing so, not only can we obtain various physical effects, i.e., nonlinear Hall effect, generation of out-of-plane magnetization and circular photogalvanic effect, but also induce

significant in-plane anisotropy of these effects. Particularly, this anisotropy can be the smoking gun evidence that proves the authenticity of these effects. The last remaining mirror plane $M_{[1\bar{1}0]}$ can be readily lifted by using *higher Miller-index* orientation, such as (122), as elucidated in the Method section of the main text.

Secondly, in an ideal situation, the designed mirror plane $M_{[1\bar{1}0]}$ at the (112)-interface can persist only if the (112)-interface is perfect without any defects. We agree that any imperfections of the interface that are inevitable in the experimental process, such as miscut induced by the polishing process, interface interdiffusion, interlayer strain, etc., would impact the $M_{[1\bar{1}0]}$ mirror plane to a certain extent. **However, these imperfections happen in a rather random and uncontrollable manner.** Thus, their perturbation on the mirror plane and the associated physical effects would be reduced to a negligible level once summed over a large area, such as a Hall bar channel. This is confirmed by three pieces of evidence in our work. (1) If these experimental imperfections would break the in-plane mirror symmetry and significantly modulate interface properties, we would have already observed the nonlinear Hall effect at (001), (110), (111)-oriented interface, as these imperfections can also happen in these orientations. Clearly, this is not the case as shown in Fig. 3d of the main text. (2) If the $M_{[1\bar{1}0]}$ is significantly broken as happened in $M_{[110]}$ and $M_{[001]}$, there would also be substantial nonlinear Hall effect and current-induced magnetization when the current is flowing along the $[11\bar{1}]$ in-plane direction. This is again not the case in our work. (3) We have also characterized the interface structure over a 54 nm width by STEM (Fig. R2.2), which shows the interface is flat within a 0.5 nm fluctuation without defects occurring in an ordered manner. Therefore, based on the group theory argument and the behaviours of the physical effect at the (112)-interface, the C_s point symmetry is the most reasonable one to describe its structure.

Thirdly, regarding the zig-zag pattern at the interface, it may happen due to interface reconstruction. However, this zig-zag pattern is a co-consequence of crystallographic engineering of the interface, rather than the origin of emergent physical effects. In other words, the zig-zag pattern itself cannot induce these effects reported in our work. To support our claim, we studied (110)-oriented LaAlO₃/SrTiO₃ interface that has been reported to show the zig-zag pattern [*ref.: A. Annadi et al., Nat Commun.* **4**, 1838 (2013); *K. Gopinadhan et al., Adv. Electron. Mater.* **1**, 1500114 (2015)]. As shown in Fig. R2.2,

there are no Kerr signals with current flowing along both in-plane directions in the (110)- LaAlO₃/SrTiO₃ heterostructure, which was prepared with the same conditions as (112)-one. Moreover, (110)-LaAlO₃/SrTiO₃ interface exhibits negligible nonlinear Hall response.

At last, regarding a stray magnetic field perpendicular to the substrate produced by the zigzag pattern, we think the Reviewer refers to the work reported by K. Gopinadhan et al. [*Adv. Electron. Mater.* **1**, 1500114 (2015)]. However, as mentioned above, zig-zag itself cannot allow the generation of out-of-plane magnetization by a current, evidenced by the absence of the MOKE signal in (110)-oriented LaAlO₃/SrTiO₃ interface. Furthermore, to the best of our knowledge, there is not yet any report of observation of room temperature magnetization/magnetic field in the LaAlO₃/SrTiO₃ interface, irrespective of the orientations. Thus, the only mechanism that can generate out-of-plane magnetization over the whole conducting channel by flowing a current would be the inverse spin-galvanic effect.

Fig. R2.1 Cross-sectional HAADF-STEM image of (112)-LaAlO₃/SrTiO₃ over a 54 nm region.

Fig. R2.2 Kerr rotation θ_K as a function of current amplitude measured in directions of [001] and $[1\bar{1}0]$ of sample (110)-LaAlO₃/SrTiO₃.

Action 2.1: We have now put the HAADF-STEM image and the MOKE measurement in the SI to show the flatness of the sample interfaces. We have also added the above discussion about the interface imperfection of the mirror plane into the SI marked in red.

Question 2.2: There is no clear description of how electrical contacts were made through the LaAlO₃ layer, which is known to have high resistance.

Answer 2.2: In our work, the electrical connections are made through the wire-bonding technique. This technique has been widely accepted in the LaAlO₃/SrTiO₃ interface transport measurement experiments, as shown in the work [Lense *et al.*, *Nat. Mater.*, **22**, 5 (2023); Chen *et al.*, *Science*, **372**, 6543 (2021)].

Action 2.2: We have already put the description of electrical contact in the “electronic transport measurements” section of Method (marked in red). “In our experiments, the Hall bars and disc bars are connected to the chip carrier with aluminium wires penetrating through the LaAlO₃ thin film using the ultrasonic wedge-bonding technique.” No further action is taken here.

Question 2.3: The mobility should also exhibit anisotropy, but this is not discussed.

Answer 2.3: We agree with the reviewer that the carrier mobility should have an anisotropic distribution as the resistivity/conductivity does. Fig. 3b of the main text shows that the conductivity/resistivity of the (112)-oriented LaAlO₃/SrTiO₃ interface possesses substantial in-plane anisotropy. As we know, the conductivity σ is determined by carrier density n and carrier mobility μ , i.e., $\sigma = ne\mu$. Since the carrier density n is a scalar quantity in the system, the only origin of the conductivity anisotropy is mobility.

To confirm the mobility anisotropy and its temperature dependence, we conducted the linear Hall measurements in a wide temperature range. As shown in the Fig. R2.3, the mobility in direction $[1\bar{1}0]$ is higher than that in the direction $[11\bar{1}]$, especially in the low-temperature range, which is consistent with our resistivity result.

Fig. R2.3 (a) Carrier mobility measured in direction $[1\bar{1}0]$ and $[1\bar{1}\bar{1}]$. (b) carrier mobility anisotropy $u_s^{[1\bar{1}\bar{1}]} / u_s^{[1\bar{1}0]}$.

Action 2.3: We have added the anisotropic mobility in SI and related description.

Question 2.4: The definition of theta is unclear, so I hope the authors specify its reference direction. Additionally, the term “LPGE” (linear photogalvanic effect) is not defined. What is its physical origin in this system? In Figure 2b, the fit is poor. At around 160 degrees, the experimental data deviate significantly from the fitted curve. In Figure 2c, the CPGE component appears absent. The red fit line diverges considerably from the experimental data.

Answer 2.4: These questions are related to the photogalvanic effect, we answer here together.

First, regarding the linear photogalvanic effect, it refers to the generation of a photocurrent under uniform illumination in a non-centrosymmetric material. In some literature, it is called the bulk photovoltaic effect [Z. Dai & A. M. Rappe, *Chem. Phys. Rev.* 4, 011303 (2023)]. The microscopic origin of the photogalvanic effect is the asymmetric momentum distribution of electrons and holes in space due to the shift current and the asymmetric scattering between photon, phonon and electrons in the non-centrosymmetric crystal [Belinicher et al., *Sov. Phys. Usp.* 23, 199 (1980)]. In contrast to the circular photogalvanic effect that only functions in the gyrotropic material and is sensitive to the mirror symmetry, the linear photogalvanic effect exists in any kind of non-centrosymmetric material.

Second, regarding the experiment data and their fittings, we think the reviewer might

misunderstand these curves in Fig. 2b and 2c. The experiment data (labelled as Exp. in the figure) comprised of both LPGE and CPGE components are marked as grey dots in the figures. The experimental data is then fitted by the curve labelled as ‘Total fit’ (see methods and Fig. R2.4). The blue (LPGE) and red (CPGE) curves shown in the figure are the components derived from the fitting (a similar fitting method is given in [Duan et al., *Nat. Nanotechnol.* 18, 867 (2023); Knoche et al., *Nat Commun* 12, 282 (2021)]. We have respectively drawn the experimental data, their fitting and the derived components of LPGE and CPGE in Fig. R2.4. Clearly, the fitting on experimental data is good.

Third, regarding the definition θ^{QWP} . θ^{QWP} is the angle between the nominal fast axis of the QWP and the polarization direction of the incident linear polarized light [*Nat Commun* 12, 282 (2021)].

Fig. R2.4 photocurrent as a function of QWP. (a) Experimental data and its total fit. (b) The LPGE component of the total fit. (c) The CPGE component of the total fit.

Action 2.4: We have now added a reference for the definition of the linear photogalvanic effect. We have also added the description of the reference direction of θ^{QWP} in the Methods. In addition, in order to make the figure clearer, we have removed the LPGE and CPGE fitting curves in Fig. 2b and 2c; instead, we have put relevant description in the main text.

Question 2.5: Figure 3f, which is referenced in line 194, appears to be missing from the manuscript.

Answer 2.5: We thank the reviewer for pointing out this typo. Figure 3f is Figure 3d.

Action 2.5: We have now corrected Figure 3f in the sentence to Figure 3d.

Question 2.6: The carrier concentration is not discussed. Can the position of the Fermi level account for the carrier density observed in the sample?

Answer 2.6: The Fermi level position is highly related to the carrier density. The carrier density of our sample, as presented in Fig. 2a in the main text, decreases with the temperature and saturates in the low temperature limit, with its magnitude in the order of $10^{13} - 10^{14} \text{ cm}^{-2}$ in the whole temperature range. These features are similar to other reported LAO/STO systems [Lesne *et al.*, *Nat. Mater.* 22, 576 (2023); Herranz *et al.*, *Sci Rep* 2, 758 (2012)]. In order to unveil the relationship between the carrier density and the Fermi level shift, we conducted the DFT calculations, as shown in Fig. R2.5. At the pristine Fermi energy, the carrier density is approximately $2.6 \times 10^{14} \text{ cm}^{-2}$. As the Fermi level is raised by +0.1 eV, the carrier density increases to $5.3 \times 10^{14} \text{ cm}^{-2}$; conversely, lowering the Fermi level by -0.1 eV results in a reduced density of $1.1 \times 10^{14} \text{ cm}^{-2}$. Although other effects might also affect the carrier density, the present calculations imply a strong connection between the Fermi level and carrier density. We also would like to address here that the calculated carrier density is based on the ideal model at zero temperature and might diverge from the real situation, as the experimentally observed carrier density at low temperature (such as 2 K) is usually at the order of 10^{13} cm^{-2} .

Fig. R2.5 Calculated 2D carrier density versus Fermi level shift in the (112)-LaAlO₃/SrTiO₃ heterostructure.

Action 2.6: we have now put the discussion of the carrier density in the supplementary information.

Response to Reviewer #3:

In their manuscript titled “Magnetization generation and giant nonlinear transport at symmetry-engineered interfaces via crystallographic orientation”, the authors H.-B. Zhang et al. present a series of experimental results in high-Miller-index LaAlO₃/SrTiO₃(hkl) heterostructures which is argued to derive from the specific nonvanishing Berry curvature and Berry curvature dipole densities hosted in the systems under investigation. The overall set of experiments and symmetry-based phenomenological models are consistent with the low Cs point-group of their heterointerfaces hosting the 2DEGs.

Question 3.1: *However, there are a number of experimental details which are missing, some desirable more detailed discussions regarding e.g., the peculiar temperature-dependence of their nonlinear Hall effect (NLHE), or with regards to the magnitude of the MOKE and the NLHE at different (11X) interfaces; and possibly a demonstration that the systems under investigation are gate-tuneable 2DEGs.*

Answer 3.1: We appreciate the Reviewer’s recognition of the significance of our work. Below, we will address all the issues raised by the Reviewer in the response letter to make the message in this manuscript clearer.

Question 3.1.1. *It would be useful/desirable to present In Fig.1, layer-resolved effective charges, in the ionic limit, of unreconstructed (11X)-LAO/STO heterostructures as in, e.g., the seminal work of N. Nakagawa et al. [Nature Materials 5, 204–209 (2006). <https://doi.org/10.1038/nmat1569>] in (001)-LAO/STO, or that of G. Herranz et al. [Scientific Reports 2, 758 (2012). <https://doi.org/10.1038/srep00758>] on (111) and (110)-LAO/STO interfaces. Given the (111)-LAO/STO interface is polar, I would assume that the (11X) interfaces are as well, but I did not “do the math”.*

Question 3.1.2. *Problematically, it seems to me that the authors argue the existence of a polar interface - as a result of the potential gradient at the interface - and cite Ref. 30 [G. Singh-Bhalla et al., Nat. Phys. 7, 80-86 (2011).], c.f. lines 83-85. Without getting into the whole debate of the “polar catastrophe scenario”, this is near complete inversion of the roles of polar interfaces and existence of 2DEGs therein. The polar interface pre-exists the 2DEG. And in virtue of the electrostatic boundary condition (in a self-consistent Poisson-Schrödinger picture), there is no further*

confinement potential to host the 2DEG without additional charges: meaning one needs electron donors, besides the band-bending due to energy mismatched conduction and valence bands at LAO/STO interfaces. More appropriate References would instead include [W.-J. Son et al., Phys. Rev. B 79, 245411 (2009)] and [P. Delugas et al., Phys. Rev. Lett. 106, 166807 (2011)].

Moreover, it is my understanding that C. Cazorla & M. Stengel [Phys. Rev. B 85, 075426 (2012)] have since shown that for metal-capped – stoichiometric – LAO/STO systems, the LAO internal polar field is a decreasing function of LAO film’s thickness, and depends on the electrostatic boundary conditions; or more precisely on the areal carrier density at both M/LAO and LAO/STO interfaces. Such that there is and I quote from them “no intrinsic built-in LAO electric field exists in the short-circuited Pt/LAO/STO capacitor system [...]” which “[...] in practice rules out the hypothesis of Zener breakdown, which was used in Ref. [G. Singh-Bhalla et al., Nat. Phys. 7, 80-86 (2011).]” However if the authors want to argue instead that the pristine interfaces are polar, and that interfacial polarity arises from the existence of an electrostatic confinement potential, then further discussions are needed.

Answer 3.1.1 and 3.1.2: Since both questions are related to the polarity and formation of the conductive interface, we would like to respond here in a comprehensive way.

Firstly, we would like to thank the Reviewer for bringing the “interface polarity” into further discussion and for the suggestion to revise the references. The polar nature of the LAO/STO interface is indeed an essential ingredient for all the physical effects discussed in our manuscript, which deserves further discussion. All in all, we agree with the Reviewer’s comment here.

It is worth noting that the “*polar conductive interface*” studied in our work can be defined as “*a conductive interface with electrical carriers experiencing an electrical potential with an asymmetrical distribution, i.e., a potential gradient*”. This polarity of the conductive interface activates the Rashba spin-orbit coupling and/or Rashba orbital-orbit coupling, underpinning all the exotic physical effect in our work. In our opinion, such an asymmetrical potential distribution (i.e. polarity) is not unusual in the conductive interface in heterostructures. This is because, by the definition of “heterostructure”, these two component layers have different chemical elements and thus, chemical potential. In the case of the LaAlO₃/SrTiO₃ interface, the electrical

carriers are confined in a narrow interface region of the SrTiO₃ side with no carriers in the insulating LaAlO₃ capping layer and the bulk of the SrTiO₃ substrate. Therefore, the distribution of electron density and the electrical potential are asymmetrical, i.e., polar. As pointed out by the Reviewer, this has been theoretically discussed by W.-J. Son et al., [Son, et al., PRB 79, 245411 (2009)] and P. Delugas et al., [Delugas et al., PRL 106, 166807 (2011)]. Also, it has been experimentally analyzed by J. Gabel et al. [Gabel et al., PRB 108, 045125 (2023)].

In this sense, the “polar interface” discussed here is not only different but also more general than that related to the “*polar catastrophe scenario*”. In other words, any conductive interface with asymmetrical potential distribution can be treated as a “polar interface”, including but not limited to those induced by the “*polar catastrophe*”. For example, the conduction layer induced in the field effect transistor can also be classified as the polar interface. To support this, we fabricated a (112)-orientated metal-oxide field-effect transistor consisting of a conductive La_{0.3}Sr_{0.7}TiO₃ layer and a SrTiO₃ insulating layer, as illustrated in Fig. R3.1a. The oxide layers of this structure are fabricated at 600 °C at a pressure of 0.01 mbar to exclude the introduction of oxygen vacancies in the SrTiO₃ substrate or the SrTiO₃ capping layer. The electron-doped La_{0.3}Sr_{0.7}TiO₃ layer was about 10 nm and shows metallic conduction. When a voltage of 0.4 V is applied to the 10 nm-SrTiO₃ gating layer, electrons in the La_{0.3}Sr_{0.7}TiO₃ layer are attracted to the interface, enhancing the asymmetry of carrier distribution. As a result, a sizable nonlinear Hall effect has been then observed under this condition, with its magnitude 10 times larger than that without the applied voltage (Fig. R3.1b). This case suffices the argument that a potential gradient is a general cause of the polarity, that are the origin of the physical effects discovered in the manuscript.

Fig. R3.1 (a) The (112)-orientated metal-oxide-semiconductor field effect transistors, La_{0.3}Sr_{0.7}TiO₃ layer is about 10 nm. (b) demonstrates its corresponding NLHE effects measured in the direction [1 $\bar{1}$ 0] with

and without a bias.

Regarding question 3.1.1, we show in Fig. R3.2 the layer-resolved effective charge distribution across the LaAlO₃/SrTiO₃ interface. It shows a similar profile as the (110)-LaAlO₃/SrTiO₃ system reported [*Herranz et al. Scientific Reports 2, 758 (2012)*; *Annadi et al., Nat Commun. 4, 1838 (2013)*]. Based on this simple ionic schematic, the ‘polar discontinuity’ does not exist. However, interfaces are conductive with a metallic behaviour till low temperatures. Thus, our result suggests that a simple catastrophe model cannot explain the formation of the 2DEG at the interface.

Fig. R3.2 Layout of ‘polar catastrophe’ model for (112) interfaces between LaAlO₃ and SrTiO₃.

Action 3.1.1 and 3.1.2: Further discussion of the interface polar nature has been added to the main text, marked as red. Fig. R3.1 and R3.2 have been added to the Supplementary Information. Regarding the reference, we would like to replace “*G. Singh-Bhalla et al., Nat. Phys. 7, 80-86 (2011)*” with the latest “*J. Gabel et al., Phys. Rev. B 108, 045125 (2023)*” and the recommended reference “[*W.-J. Son et al., Phys. Rev. B 79, 245411 (2009)*; *P. Delugas et al., Phys. Rev. Lett. 106, 166807 (2011)*].”

Question 3.1.3: Line 113 “conventional cases” is not well defined and could be replaced by “canonical low-Miller-index LAO/STO interfaces” which I believe is what is implied. This, Ref. [29] is here maybe not the most relevant reference.

Answer 3.1.3: Physical properties of LaAlO₃/SrTiO₃ with orientations of (001), (011) and (111) have already been investigated by the community, while (112)-LaAlO₃/SrTiO₃ is a new member. This is the reason why we initially defined (112) as the ‘unconventional’ case in the LaAlO₃/SrTiO₃ family, while others are considered ‘conventional’. We thank the reviewer for pointing this out, and we agree that

‘canonical low-Miller-index $\text{LaAlO}_3/\text{SrTiO}_3$ interfaces’ is a more proper phrase to describe (011) and (111)- $\text{LaAlO}_3/\text{SrTiO}_3$ interfaces in our work. Regarding the reference, ref. [29] is a work studying the [111]- $\text{LaAlO}_3/\text{SrTiO}_3$ interface, which fits the picture.

Action 3.1.3: We have now changed the phrase accordingly

Question 3.2

Question 3.2.1. The authors present in the inset of Fig. 2a, the temperature-dependence of the electronic mobility and carrier density in their 112-Lao/STO interface. It would be good to specify if those measurements were conducted in Hall bar devices; as well as disclose the raw data of their Hall effect measurements (in Extended data?), as this is the first ever report of transport properties at this specific interface.

Question 3.2.2: This open the questions of whether the Hall effect purely linear? And at which magnetic field value was the Hall coefficient estimated to calculate the sheet carrier densities?

Answer 3.2.1 & 3.2.2: These two questions are related to the Hall measurement. Regarding the data shown in the inset of Fig. 2a, they were obtained in a Hall bar device with the magnetic field sweep between 3 T and -3 T, as shown in Fig. R3.3a. The carrier density and mobility are all derived from the linear fits of these curves.

To further check the linear dependence in these ordinary Hall effects, we conducted the Hall measurement with the magnetic field applied as large as ± 8 T. As demonstrated in Fig. R3.3b, the linearity of these curves at all temperature ranges is maintained up to ± 8 T. Therefore, the magnetic field of ± 3 T is sufficient for the Hall measurement to investigate the carrier density and mobility.

Fig. R3.3 (a) Hall measurement in the magnetic field up to ± 3 T. (b) Hall measurement in the magnetic field up to ± 8 T.

Action 3.2.1 & 3.2.2: We have put a statement in the SI to declare that the carrier mobility was obtained from a Hall measurement in the application of ± 3 T.

Question 3.2.3: *Have the authors attempted to perform a gate-tuning of the metallicity of the 2DEG at low T (below 10K), where the extremely large dielectric constant of the STO substrate should allow for an efficient electrostatic field-effect in back-gating geometry. This is possibly beyond the scope of the present work, but would represent a substantial addition to the current manuscript.*

#3.2.4. *Obviously, the corollary to the latter becomes how does the NLHE evolves as a function of sheet conductance (or back-gate voltage)?*

Answer 3.2.3 & 3.2.4: Indeed, the gate tunability of various physical effects is a merit of the LaAlO₃/SrTiO₃ interface, especially at low temperature due to the substantially increased dielectric permittivity of the SrTiO₃ crystal.

As suggested by the Reviewer, we present here the results of the gate-tuning of the conductivity and NLHE at 1.55 K. To enable a large field effect, we have thinned the thickness of the STO substrate to ~ 50 μm . When sweeping the back gate voltage from positive to negative, the conductivity decreases monotonically by 1 order of magnitude. Meanwhile, the NLHE coefficient firstly increases and then decreases with the back-gating voltage, showing a variation over 10-fold (see Fig. R3.4). This reveals the highly tunability of the nonlinear transport of the symmetry-engineered LaAlO₃/SrTiO₃ heterostructure. Nevertheless, elaboration of such a gate dependence is non-trivial and

challenging. The field effect on the LaAlO₃/SrTiO₃ heterostructure affects various aspects of the interfaces, including potential asymmetry, carrier distribution, Fermi level, orbital occupation, etc. We believe it is beyond the scope of the present manuscript and we would like to study it in detail in a following work.

Fig. R3.4 Gate-tuning of conductivity and NLHE.

Action 3.2.3 & 3.2.4: We believe current manuscript is already a complete piece of work and we would like to study the gate tunability of this system comprehensively in a separate work. We prefer to publish Fig. R3.4 separately.

Question 3.3

Question 3.3.1. *The short discussion (lines 107-109) related to a small upturn in R_s “commonly existing in LAO/STO 2DEG systems” should be slightly moderated. In some sense, yes, it is commonly observed, but somewhat not in state-of-the-art, clean and optimally doped systems [c.f. G. Herranz et al., Phys. Rev. Lett. 98, 216803 (2007)]. The upturn is, without debate, a sign of increased scattering rate of electrons at low T . The underlying (microscopic) mechanism is much more difficult to pinpoint. This is generally difficult due general tendency to preferentially form either cationic (La, Al) or anionic/oxygen vacancies during the growth of the LAO layer, depending on the partial pressure of oxygen.*

I would certainly not put “Kondo” and “commonly” side-by-side. Despite the speculated role of oxygen deficiency clusters and their associated magnetism in STO-

based systems, this is highly debated in the community; see also the decades old controversy of magnetism in d^0 (or $d^0+\delta$) electron systems. The authors should provide a couple more references for the upturn when the growth conditions are suboptimal, such as in the early reports by A. Brinkman et al. [Nature Materials 6, 493–496 (2007). <https://doi.org/10.1038/nmat1931>], or other tentative explanation based on Kondo-screening [J. Ruhman et al., Phys. Rev. B 90, 125123 (2014) <https://doi.org/10.1103/PhysRevB.90.125123>]

On the other hand, in the seminal work of G. Herranz et al. [Scientific Reports 2, 758 (2012). <https://doi.org/10.1038/srep00758>], on (111) and (110)-LAO/STO interfaces, the upturn was very pronounced for thick (111)- and (110)- interfaces, as opposed to nearly absent for thinner LAO layers. However, in later reports, this can be entirely mitigated by finding optimal growth conditions, and this thickness-dependent upturn is no longer seen as an intrinsic property of the interface.

Answer 3.3.1: We appreciate the Reviewer’s profound insight into the low-temperature transport behavior of the $\text{LaAlO}_3/\text{SrTiO}_3$ interface. We are very grateful that the reviewer points out this improper phrase in our manuscript. To avoid misinterpretation to the readers, we will remove the ‘Kondo effect’ in the sentence as in this manuscript. Since we are mainly focusing on the room temperature properties of the (112)- $\text{LaAlO}_3/\text{SrTiO}_3$ heterostructure, further detailed analysis and optimization of this upturn are beyond the scope of this work.

Action 3.3.1: We have revised the sentence to ‘The small upturn of R_s at lower temperatures indicates the enhanced electron scattering at low temperatures.’ In addition, we also put the recommended references [Brinkman et al., Nature Materials 6, 493 (2007); Herranz et al., Phys. Rev. Lett. 98, 216803 (2007)] along with the statement.

This brings me to the following questions.

Question 3.3.2. *What is the thickness of each of the LAO (hkl) layers studied in this work?*

Question 3.3.3. *In Extended Data Fig.1, the authors present a typical X-Ray diffraction*

pattern for the (112) interface. How about all the other interfaces? Namely the LAO/STO (113), (114), (115), as well as the (110), (111) and (001) interfaces; all of which are discussed in relation to the NLHE displayed in Fig. 2d.

Answer 3.3.2 & 3.3.3: These two questions are related to the film thickness and XRD results, we answer these two questions together. The (112)-LaAlO₃/SrTiO₃ thin film studied in the work was grown by PLD with a thickness of ~4 nm, as determined by STEM (see Fig. R2.1). Other (hkl)-LAO/STO thin films (for NLHE studies) were grown in the same condition and thus have similar thickness (~4 nm). Their XRD spectroscopies are shown in Fig. R3.5. Clearly, (112), (001), (110), (111)-oriented LAO/STO heterostructures all show the interference fringes of the LaAlO₃ peaks, indicating high film quality. In contrast, the fringes are absent in the (113) and (114) oriented heterostructures. This also implies that the (112)-orientation is the optimal high-miller index orientation to explore. In addition, we also characterized the surface topography of all these orientations as shown in Fig. 3.6. All of them show ultra-smooth surfaces with sub-nanometre roughness.

Fig. R3.5 XRD studies of (hkl)-LaAlO₃/SrTiO₃ thin films. (a) (001)-LaAlO₃/SrTiO₃. (b) (110)-LaAlO₃/SrTiO₃. (c) (111)-LaAlO₃/SrTiO₃. (d) (112)-LaAlO₃/SrTiO₃. (e) (113)-LaAlO₃/SrTiO₃. (f) (114)-LaAlO₃/SrTiO₃.

Fig. R3.6 Topography images of (hkl)-LaAlO₃/SrTiO₃ thin films. (a) (001)-LaAlO₃/SrTiO₃ (b) (110)-LaAlO₃/SrTiO₃ (c) (111)-LaAlO₃/SrTiO₃ (d) (112)-LaAlO₃/SrTiO₃ (e) (113)-LaAlO₃/SrTiO₃. (f) (114)-LaAlO₃/SrTiO₃.

Action 3.3.2 & 3.3.3: We have now put a sentence to elaborate on the thickness of the sample in the methods section. Fig. R3.5 and Fig. R3.6 have been added to the SI.

Question 3.4

Regarding the magneto-optical Kerr effect (MOKE) data and the circular photogalvanic effect (CPGE) presented in Figure 2.

Question 3.4.1. *I understand that, on one hand, the current-induced MOKE signal could be expected to be small. However, on the other hand, the sensitivity of (longitudinal) MOKE setups can reach down to a couple of nrad, so the signal reported by the authors is rather large in comparison. This brings me to the following: How come the Kerr rotation angle at the lowest probed current density (as well as throughout the whole [11-1] data set) is as large as 0.05 microrad (or 50 nrad)? Can the authors comment on the tentative origin of this relatively large offset value?*

Answer 3.4.1: The sensitivity of the longitudinal MOKE in previous works can reach ~10 nrad [e.g. Choi et al., Nature 619, 52 (2023)], which is about 5 times better than our work (~50 nrad). To achieve the state-of-the-art sensitivity, each part of the MOKE system, especially power and polarization of the laser, and the temperature of optical components used in the light path, has to maintain as stable as possible. In our system,

we have designed a home-made system to maintain the temperature fluctuation of optical components, including the waveplates and the Wollaston prism, within ± 1 mK. However, the He-Ne laser used in our system exhibits the sizable fluctuation of power and polarization, which are difficult to suppress with our current technique and limited funding. This fluctuation of the laser setup causes noise in our MOKE system and reduces its sensitivity. Replacing the He-Ne laser with a sophisticated Ti: sapphire laser (which is very expensive) will likely increase the MOKE sensitivity [as used in *Choi et al.*, *Nature* 619, 52 (2023)].

Regarding the ~ 50 nrad offset of the MOKE signal shown in Fig. 2d of the main text, it probably arises from three factors: (1) the finite extinction ratio (100000:1) of the Wollaston prism placed before the balanced detector; (2) the intrinsic offset/noise and finite resolution of our lock-in amplifier; (3) finite electromagnetic cross talk between different electronic components in our home designed MOKE system.

Nevertheless, our MOKE system with its present sensitivity (~ 50 nrad) is good enough to resolve the MOKE signal in the (112)-interface induced by current. More importantly, a clear contrast of the MOKE signal between $[1\bar{1}0]$ and $[11\bar{1}]$ in-plane current direction is easily observed by our system, consolidating our claim and confirm the genuineness of the phenomena. We would like to optimize our MOKE system in the future once further funding is granted.

Action 3.4.1: We have added further discussion of the MOKE signal and noise in the Methods section and SI, marked as red.

Question 3.4.2. The data in Fig.2e is presented for a current density of 2 A/m, that is, given the 10 microns cross-section of the Hall bar, for a current amplitude of 20 microAmps. This is 40 times larger than any of the second harmonic I-V characteristic presented in Fig. 3a and Extended Data Fig. 4c. For consistency, and in the spirit of transparency, the authors could either display a secondary x-axis (top of panel 2d) with the absolute value of the current.

Answer 3.4.2: To our best knowledge, it is more common to use the current density to describe the current-induced magnetization [*Choi et al.*, *Nature* 619, 52 (2023)]; *Stamm et al.*, *Phys. Rev. Lett.*, 119, 087203 (2017); *Lyalin et al.*, *Phys. Rev. Lett.*, 131, 156702

(2023);]. In order to make it consistent with other reported works, it is rational to choose A/m as the unit while not μA . But we agree with the reviewer's opinion that a secondary axis of the unit μA on the top of the figure makes the picture clearer. This question is also related to the different current ranges used in NLHE and MOKE, we will further discuss this question in Question 3.5.1&3.5.2.

Action 3.4.2: We have now put a secondary x-axis on the top of the panel.

Question 3.4.3 Line 115 the authors claim “To show that indeed the interface possesses a monoclinic C_s symmetry, we have measured the CPGE [...]”. This is incorrect and should be rephrased. The existence of a CPGE cannot prove per say the C_s symmetry of the system. The CPGE is instead a potentially symmetry-allowed effect, and its anisotropy consistent with C_s symmetry.

Answer 3.4.3: As we mentioned in Answer 1.1, the CPGE only functions in gyrotropic materials and is characterized by second order axial tensors. As a result, when light is shining along the interface/surface normal direction, only the current generated by the CPGE could flow in the direction normal to the mirror symmetrical plane. Therefore, for a 2D structure, if it has more than one mirror plane (e.g., C_{2v} or C_{3v}), then it has no CPGE; If it has no mirror plane (C_1), then it has the CPGE along all the in-plane directions. Only in the system with one mirror plane can the CPGE exhibit characteristic directional behaviours, that is, its direction is perpendicular to the only mirror plane. Therefore, the existence of the CPGE in only one direction is consistent with the C_s symmetry.

Action 3.4.3: To make our statement more rigorous, we have rephrased our sentence of Line 115 as:

“To get an insight into the mirror symmetry breaking in the designed (112)-interface, we have measured the circular photogalvanic effect (CPGE) on the same samples by illuminating the surface using a circularly polarized 405 nm laser beam with normal incidence.”

Question 3.5

Last but not least, the nonlinear Hall effect data (Fig. 3). The symmetry-based arguments made in favor of the existence of a Berry curvature dipole-driven NLHE, and the first-principle calculations associated to it, are not being questioned here.

Rather:

Question 3.5.1 In relation to my previous point: what happens to the nonlinear I-V characteristics (Fig. 3a, Ext. Data Fig. 4) when sourcing current densities comparable to those used in Fig.2? And very importantly, do those NL I-V present a hysteretic behaviour?

Question 3.5.2 . Same question goes for the longitudinal linear I-V characteristic, ideally $I(\omega)$ vs. $V_{xx}(\omega)$ (see Extended Data Fig. 4b) - and $I(\omega)$ vs. $V_{yy}(\omega)$ - in the same current density range)? In general, how can the authors rule out heating related effects or other sources of nonlinearities?

Answer 3.5.1 & 3.5.2: These two questions are related to the transport behaviours when a larger current is sourced, we answer here together.

The main reason why using different current magnitudes for NLHE and MOKE measurement is due to their contrasting detectivity. Both physical effects are characterized by using an AC driving current and detecting the output voltage signal with a lock-in amplifier. The lock-in amplifier can detect an AC voltage with an RMS amplitude as low as tens of nV. In the case of the NLHE measurement, the second order nonlinear Hall voltage sign can easily reach several μV with an input a.c. current of a few hundred nA (see Fig. 3a). However, in the case of the MOKE measurement, an input current of tens of μA is required to induce a sensible output signal from the balanced detector. This is common for the MOKE measurement on the current-induced magnetization, such as the spin Hall effect [*Stamm et al., Phys. Rev. Lett., 119, 087203 (2017)*; *Lyalin et al., Phys. Rev. Lett., 131, 156702 (2023)*].

To answer the reviewer's request on the transport properties under larger ac current, we conducted the transport measurement with a maximum 20 μA applied (see Fig. R3.7). As is seen from the figures, all three measured voltage drops do not show hysteresis behaviours. V_{xx}^ω linearly grows with I_x^ω and V_{xy}^ω keeps it small value all over the current range due to misalignment. Meanwhile, $V_{xy}^{2\omega}$ increases quadratically up to 20 μA .

Fig. R3.7 (a) The 1st order voltage drops V_{xx}^{ω} and V_{xy}^{ω} and (b) the 2nd order voltage drop $V_{xx}^{2\omega}$ as a function of ac current up to 20 μA .

(2) Heating-related effects and other sources induced by the nonlinearity can be ruled out by two phenomena in our work: a) the significant anisotropic performance of NLHE, CPGE and MOKE signals between the two orthogonal directions. b) the above phenomena in (001)-, (011)- and (111)- $\text{LaAlO}_3/\text{SrTiO}_3$ are not observable under the same measurement condition.

Action 3.5.3: Fig. R3.7 has been added to the SI.

Question 3.5.3 In Fig. 3c, how is each data point obtained? Do the authors fit a full nonlinear I-V characteristics at a given temperature, or is it being measured at constant current bias upon continuously sweeping the temperature (cooldown? or warming? And at which rate?).

Answer 3.5.3: To obtain the temperature-dependent NLHE coefficient, we measured both V_{xx}^{ω} and $V_{xy}^{2\omega}$ continuously by heating the sample with the ac current biased (i.e., 1 μA). Then the NLHE was calculated by using the formula $E_{xy}^{2\omega}/(E_{xx}^{\omega})^2$, see Fig. 3c for example. This method has also been adapted to measure the temperature-dependent NLHE by other works where lots of data are collected in a wide temperature range [Min et al., Nat. Commun. 14, 364 (2023); Itahashi et al., Nat. Commun. 13, 1659 (2022)]. In order to further confirm the rationality of this method, we also conducted the measurement of both V_{xx}^{ω} and $V_{xy}^{2\omega}$ upon sweeping the ac current at the fixed temperature. The plot of $V_{xy}^{2\omega}$ vs $(V_{xx}^{\omega})^2$ at various temperatures is given in Fig. R3.8.

The NLHE coefficient obtained at 450 K, 410 K, 375 K, 350 K, and 300 K by fitting the linear $V_{xy}^{2\omega}$ vs $(V_{xx}^\omega)^2$ curves are 116, 100, 69, 55, 41 $\mu\text{m V}^{-1}$, respectively. These values are close to the data obtained by continuously measuring NLHE.

Fig. R3.8 The temperature-dependent NLHE measured at fixed temperatures.

Action 3.5.3: We have now put the relative description of measuring temperature-dependent NLHE in the Method section.

Question 3.5.4. *Could the authors comment further on the quite peculiar temperature-dependence of the NLHE reported in Fig. 3c? is there a tentative explanation for the strong dip at about 100 K, and for a recovery of the effect's magnitude at lower temperature? Can this be understood in relation to the measured temperature-dependence of the sheet conductance and sheet carrier density? The argument regarding the Fermi level.*

Answer 3.5.4: To understand the peculiar temperature dependence of the NLHE reported in our manuscript, we need to resolve its different contributions.

It has been reported that, in addition to the intrinsic contribution of the BCD, extrinsic contributions, including side jumping and screw scattering, play an important and even dominant role in the NLHE. To resolve various contributions, we employ the scaling method at both low (< 30 K) and high temperature (> 100 K) ranges (Fig. R3.9). The two curves in Fig. R3.9 are obtained by relating the NLHE coefficient $E_{xy}^{2\omega}/(E_{xx}^\omega)^2$ and conductivity σ_s at each temperature. According to the theory [Du et al., Nat. Commun. 10 3047 (2019)], the scaling law for NLHE can be written as: $E_{xy}^{2\omega}/(E_{xx}^\omega)^2 =$

$A_1 \left(\frac{\sigma_s}{\sigma_0}\right)^2 + A_2 \frac{\sigma_s}{\sigma_0} + A_3$, where A_1, A_2, A_3 are the scaling parameters, σ_0 is the conductivity at 1.55 K. The fitting results of A_1, A_2, A_3 are shown in Table R3.1. Based on the NLHE scaling law, the scaling parameters are expressed as:

$$A_1 = C^{sk,2}\sigma_0 + (C_{00}^{sk,1} + C_{11}^{sk,1} - C_{01}^{sk,1}) \quad (3.1)$$

$$A_2 = C_{01}^{sk,1} - 2C_{11}^{sk,1} + C_0^{sj} - C_1^{sj} \quad (3.2)$$

$$A_3 = C_{in} + C_1^{sj} + C_{11}^{sk,1} \quad (3.3)$$

where C_{in} is an intrinsic contribution, C_i^{sj} the side-jump, $C_{ij}^{sk,1}$ the Gaussian skew-scattering, $C^{sk,2}$ the non-Gaussian skew-scattering, and $i, j = 0$ (1) denotes the defects (phonon) scattering source. There are seven parameters, i.e., seven different contributions in the above equations, which are impossible to resolve quantitatively. To get a tentative insight into these contributions, here we take a bold but almost reasonable assumption.

Fig. R3.9 Scaling law analysis of the NLHE at temperature (a) lower than 30 K and (b) higher than 100 K.

Table. R3.1 Scaling parameters

	A_1	A_2	A_3
T < 30 K	7.56	-42.4	56.9
T > 100 K	90.5	-112.4	36.6

Since $C^{sk,2}$ only exist in the A_1 parameter, the non-Gaussian skew-scattering only plays a secondary role. In the temperature range below 30 K, the parameter A_1 is almost

one order of magnitude smaller than A_2 and A_3 . This indicates that the screw scattering contributions play a minor role at the low temperatures. Also, the phonon density is largely suppressed at low temperature, it would be reasonable to assume C_0^{sj} (i.e., side jump due to defects) is much larger than C_1^{sj} (i.e., side jump due to phonons). Thus, $A_2 = -42.4 \approx C_0^{sj}$. Similarly, $A_3 = 56.9 \approx C_{in}$.

In the temperature range above 100 K, the parameter A_1 shares a similar magnitude with A_2 but with opposite sign, while A_3 about three times smaller. Based on equations (2) and (3), the opposite sign between A_1 and A_2 is most likely due to the opposite sign of $C_{11}^{sk,1}$ and $C_{01}^{sk,1}$ in these two equations. Since $C_{00}^{sk,1}$ (screw scattering due to defects) only exist in A_1 , it may play a minor role here. Moreover, since the side jump process is insensitive to the temperature, we assume C_0^{sj} remains as a constant over the whole temperature range. Thus, equations (3.1-3.2) can be approximated as:

$$A_1 = C_{11}^{sk,1} - C_{01}^{sk,1} = 90.5 \quad (3.4)$$

$$A_2 = C_{01}^{sk,1} - 2C_{11}^{sk,1} - C_1^{sj} = -70 \quad (3.5)$$

$$A_3 = C_{in} + C_1^{sj} + C_{11}^{sk,1} = 36.6 \quad (3.6)$$

Thus, we get $C_{in} = 57$, which almost equal to that derived at low temperature range, supporting the above approximation.

Based on the above analysis, we can reach here a qualitative understanding of the origins and temperature dependence of the NLHE in the (112)-LAO/STO interface. Clearly, both the intrinsic and extrinsic contributions play important and compensating roles in this system. For example, the extrinsic contribution $C_1^{sj} + C_{11}^{sk,1}$ ($= -20.4$) is opposite to that induced by Berry curvature dipole ($C_{in} = 57$).

The peculiar temperature dependence can be ascribed to the role of the parameter A_2 , i.e., contributions of screw scattering and side jump. This also means that the NLHE effect of the (112)-interface is highly related to its conductivity with an approximated quartic dependence. When the temperature decreases from 300 K to 100 K, the conductivity and mobility increase, the screw scattering process is enhanced, inducing a negative contribution that compensates that of the intrinsic one. At the temperature below 30 K, the density of phonons is significantly reduced and thus, the related screw scattering contributions ($C_{01}^{sk,1}$ and $C_{11}^{sk,1}$) are suppressed. In this circumstance, the side

jump and intrinsic dominate. With further decreasing temperature towards 1.55 K, the conductivity decreases, the compensating contribution of side jump by defects is reduced, leading to an enhanced NLHE.

Action 3.5.4: We have added the relevant comments on the behaviour of the temperature-dependent NLHE in the main text and put the scaling law of the NLHE in SI.

Question 3.6 Methods section:

(1) Line 324: annealed in which conditions? In vacuo in-situ? in air? In pure oxygen flow furnace? (2) Are all the (11X), the (001), the (110) and (111) interfaces grown in the same conditions?

Answer 3.6: Substrates are annealed in air at 1000 °C for 2 hours; The thin film growth conditions for all (hkl)-LaAlO₃/SrTiO₃ thin films are the same.

Action 3.6: We have put a relative description in the Methods section.

Question 3.7 Miscellaneous:

Question 3.7.1 *In Fig. 3a, is the y-axis not going from negative to positive values? Or is this deliberate?*

Answer 3.7.1: We thank the reviewer for pointing this out. Initially, we used the phase to denote the ‘negativity’. However, we notice that it is more common to set the y-axis from negative to positive, such as in the reference [Kang et al., Nat. Mater. 18, 324 (2019)].

Action 3.7.1: We have changed the y-axis.

Question 3.7.2 *In Fig. 1b: if I am not mistaken, the “x”-vector: [11-1] should point towards the bottom of the page, otherwise the arrow actually indicates the [-11-1] direction.*

Answer 3.7.2: We are very grateful that the reviewer points this out. The arrow of the

direction $[11\bar{1}]$ in the figure indeed shall point downwards.

Action 3.7.2: We have correspondingly revised the coordinate in the figure.

Question 3.7.3 Line 34: typo “Wely” instead of “Weyl”.

Answer 3.7.3: We thank the reviewer for pointing out this typo.

Action 3.7.3: We have corrected this typo.

Question 3.7.4 Line 82: “ultrasmooth” could be replaced by “atomically (?) sharp”, though the lack of spatially-resolved element specific EDXS (energy dispersive x-ray spectroscopy) may prevent making that claim.

Answer 3.7.4: In the main text, we describe the surface as ‘ultrasmooth’ but not the interface. Thus, we believe this description is correct as the AFM data clearly show the roughness of the sample is only about 90 pm. Since the reviewer mentioned the interface, from the HAADF-STEM image we can tell that the interface between LAO-STO is seemly sharp. However, we understand that it is usually hard to achieve a perfect sharp interface where the mixture of atoms might happen [*Nakagawa et al., Nature Mater* 5, 204 (2006)].

Action 3.7.4: No action taken.

Question 3.7.5 Line 181: “it exists”, “it” should be deleted here.

Answer 3.7.5: We thank the reviewer point out this grammar problem.

Action 3.7.5: we have removed ‘it exists’ from the sentence.

Question 3.7.6 Line 229 and 230: “Fermi lines” (twice) could be replaced with “Fermi level” or “Fermi energy” (and give its corresponding definition).

Answer 3.7.6: We believe the term ‘Fermi lines’ is correct here. According to the definition, in the Brillouin zone (BZ) of a two-dimensional (2D) material, Fermi lines represent the constant-energy contours (Fermi contours) where the electronic band energy equals the Fermi level. However, Fermi level refers to the scalar energy value.

In the content of calculating Berry curvature or Berry curvature dipole distribution over the Brillouin zone, it is correct to use ‘Fermi lines’ [*Mercaldo et al., npj Quantum Mater.* 8, 12 (2023); *Lesne et al., Nat. Mater.* 22, 576 (2023)].

Action 3.7.6: No action taken.

Question 3.7.7 Line 230: “exhibit symmetries concerning both $k_x = 0$ and $k_y = 0$ ”: which symmetries? What is meant here by “concerning”? I am confused.

Answer 3.7.7: Initially, we wanted to express that the D_{yz} is symmetric about $k_x = 0$ and $k_y = 0$, as displayed in Fig. 4c. We understand that the word ‘concerning’ is not proper here, which might cause confusion as the reviewer mentioned.

Action 3.7.7: We have changed ‘concerning’ to ‘with respect to’.

Question 3.7.8 Line 250: “are” should be “were” or “have been” (due to the use of before at the end of the sentence).

Answer 3.7.8: We thank the reviewer for pointing this out.

Action 3.7.8: We have revised the sentence.

Question 3.7.9 Extended Data Fig. 7, what are the corresponding current densities?

Answer 3.7.9: The corresponding currents in these measurements are in the range of 1 μA . The feature of the quadratic dependence of $V_{xy}^{2\omega}$ on I_x^ω (or V_{xx}^ω) is clearly shown when the applied current reaches 1 μA , thus it is rational to choose this range.

Action 3.7.9: We have put the current amplitude in the figure captions.

Question 3.7.10 On many occasions, articles such as “the” or “a/an” are missing, or on the contrary not necessary. I would urge the authors to take the necessary English language revisions.

Answer 3.7.10: We thank the reviewer for carefully checking the grammar of this manuscript.

Action 3.7.10: We have correspondingly revised the mentioned typos and also checked the manuscript thoroughly.

***Question 3.8:** In conclusion, I cannot recommend the present manuscript, as it is, for publication in Nature Communications, but I encourage the authors to address my comments and questions. Upon satisfying revisions, I am of the opinion that this work would be of great interest to the broader solid-state physics community with a keen interest on nonlinear transport, and their consequences for applications, in low-symmetry systems.*

Answer 3.8: We appreciate the reviewer's cognition of our work and thank the reviewer for giving us many valuable suggestions. By answering the comments point-to-point, we believe our manuscript has now reached the standard of Nature Communications.

Response Letter Version 2

We would like to thank the Reviewers for their valuable comments and appreciations of our work. We have accordingly revised the manuscript and the supporting information. We believe we have positively addressed all comments made by the reviewers. A point-by-point response to the reviewers' comments and changes made in the revised manuscript are given below.

Response to Reviewer #1:

Question 1: I have carefully studied the authors' response letter and revised manuscript, which have addressed my concerns. I would like to recommend the revised manuscript for publication.

Answer 1: We are very grateful that the reviewer gave constructive suggestions to make this manuscript more clear and solid. We also thank the reviewer for the recognition of our work.

Action 1: No action taken.

Response to Reviewer #2:

I would first like to thank the authors for their efforts in revising the manuscript and for addressing several of the previous concerns. While some points have indeed been clarified, there remain a number of important issues that are still insufficiently resolved:

Answer: We thank the reviewer for giving us important suggestions and pointing out some remaining issues. Below, we address in detail all the raised questions and take the corresponding actions.

Question 2.1: Role of defects and interface structures: The authors argue that if experimental imperfections were to break the in-plane mirror symmetry and significantly alter the interface properties, nonlinear Hall effects should also be observable in the (001), (110), and (111) interfaces, which is not the case (as shown in Fig. 3d of the main text). However, one should keep in mind that the nature of defects

and interdiffusion strongly depends on the growth orientations. In high-energy surfaces such as (112), different kinds of interface reconstructions are likely to occur compared to (001), (110), or (111). Although many effects may indeed average out, not all of them do. Since the manuscript does not discuss the difference in defect formation among these orientations, it is still possible that the observed effects originate from specific microstructural features unique to the (112) interface. Changes in oxygen octahedral rotations should also be considered. For example, in systems such as SRO, out-of-plane spin components have been predicted as a consequence of octahedral rotations. Therefore, the absence of nonlinear Hall effects or current-induced magnetization in (001), (110), and (111) interfaces does not necessarily prove that the observed effects in (112) originate solely from the preservation of the $M_{[1-10]}$ mirror plane.

Answer 2.1:

We respectfully disagree with the Reviewer's interpretation, especially with regard to the concluding statement. The development of nonlinear transport and current-induced magnetization does NOT originate from the '*preservation*' of the mirror symmetry. Rather, it is a direct consequence of the '*breaking*' of the other mirror planes, such as $M_{\{100\}}$. This interpretation is fully consistent with Curie's principle, which states that "asymmetry is what creates a phenomenon". The deliberate preservation of the $M_{[1\bar{1}0]}$ mirror plane serves exactly this purpose: to *prevent* the onset of nonlinear transport effect with input current flowing along the mirror direction (i.e., $[11\bar{1}]$ in our work). Thus, breaking of all the other mirror planes except the $M_{[1\bar{1}0]}$ provides evidence, what we term "smoking gun confirmation", for our central hypothesis, specifically the creation of the in-plane anisotropy and observing the associated effects in the orientation-engineered interfaces. Such in-plane anisotropy has been systematically and repeatedly validated by our experimental results.

When the final mirror plane $M_{[1\bar{1}0]}$ is broken (for example, in the (122)-LAO/STO orientation), no mirror plane is retained at the interface. As a result, nonlinear transport properties are symmetry-allowed in all in-plane directions. By contrast, systems with more than two planes with preserved mirror symmetries, such as the (110)-, (100)- and (111)-LAO/STO, impose mutual constraints that forbid the emergence of a nonlinear Hall effect. This has been rigorously established in the Method section with explicit

mathematic support. Thus, the experimental observations reported here are consistent with the symmetry analysis.

Regarding the role of defects and interface structures, we would like to discuss it in two aspects.

(i) Firstly, we concur with the Reviewer that “nature of defects and interdiffusion strongly depends on the growth orientations”. Importantly, all interfaces reported in this work were grown under the same condition. Therefore, any orientation-dependent variation in defect configuration or interdiffusion, if present, must be attributed solely to the crystallographic orientation, precisely as the Reviewer has suggested. In this sense, the “*special*” defects and interdiffusion in the (112)-orientation, as well as their role in the observed effects, if any, should be an intrinsic outcome of our symmetry-engineering strategy. In other words, such defect-mediated effects are not extraneous but rather an inseparable component of the mechanism we deliberately exploit.

Since the polar catastrophe scenario is unlikely to operate in the (112)-orientation, oxygen vacancies are expected to dominate the interface conduction, as already discussed in our previous response letter. Consequently, defects indeed play an important role in nonlinear transport and magnetization generation at the (112)-interface, but in a distinct way as the Reviewer suggests here. In our previous response letter ‘Answer 1.4’, we have discussed in detail the role of defects in the nonlinear Hall effect (NLHE), interrogating the scaling law by varying the temperature and thus the conductivity. These analyses demonstrate that the effects associated with defect scattering, such as the side jump, skew scattering, coexist with the intrinsic Berry curvature dipole contribution. **The overall transport property is the sum of asymmetric scattering and intrinsic properties, both of remain strictly governed by the underlying symmetry constraint.** This is more clearly seen from the tensor properties of the NLHE and the anisotropic resistive response as a function the applied current direction.

2. Secondly, we address the Reviewer’s point regarding the possibility of “different kinds of interface reconstructions” in (112)-oriented interface and whether “such a unique interface reconstruction” is the fundamental origin of the observed effects or not.

The direct characterization of defects and interface reconstruction in complex oxide heterostructures, as well as disentangling their precise role in electronic transport, is

technically demanding and could itself constitute an independent research project. Nevertheless, recognizing the importance of this issue, we have designed three contrast experiments: (b1) (112)-Pt/SrTiO₃/La_{0.3}SrTiO₃ field effect transistor grown on SrTiO₃ substrate (also seen in the first response letter ‘Answer 3.1’) and (b2) reduced (112)-SrTiO₃ substrate.

Before presenting the detailed results of these three experiments, we summarize the main conclusion: (1) Nonlinear Hall effect **is consistently observed whenever (112)-oriented interfaces are formed, regardless of the different types/natures of the defects** (see La_{0.3}Sr_{0.7}TiO₃ field effect transistor). (2) **Defects alone cannot generate nonlinear transport in the absence of symmetry breaking.**

These findings strongly support our interpretation that the NLHE originates fundamentally from symmetry considerations rather than being solely a by product of defect-driven interface reconstructions.

We are now presenting the detailed results of the three contrast experiments.

(b-1) La_{0.3}Sr_{0.7}TiO₃ field effect transistor: the (112)-oriented La_{0.3}Sr_{0.7}TiO₃/SrTiO₃ bilayer were grown under high oxygen pressure and relatively low temperature (i.e., 600 °C and 0.01 mbar oxygen pressure) to minimize the oxygen vacancy density.

In this system, both the SrTiO₃ substrate and SrTiO₃ film are insulating, with the conduction confined within a 10 nm La_{0.3}SrTiO₃ thin film. As shown in Fig. RR2.1c, the pristine bilayer already shows a small nonlinear Hall effect at room temperature without apply a gate field. This residual response most likely arises from inequivalent boundary conditions at the bottom interface of La_{0.3}Sr_{0.7}TiO₃/SrTiO₃ substrate and SrTiO₃/La_{0.3}Sr_{0.7}TiO₃ top interface. When a gate voltage of 0.4 V is applied via top SrTiO₃ layer, the NLHE increases significantly. Crucially, the nature and configuration of the La_{0.3}Sr_{0.7}TiO₃ transistor are entirely distinct from the LaAlO₃/SrTiO₃ interface, with the exception of the crystallographic orientation. The persistence and enhancing of NLHE in this system therefore unambiguously demonstrate that the phenomenon originates from orientation-induced mirror-symmetry breaking, rather than from any particular defect configurations.

Fig. R2.1 NLHE study in a field effect transistor (112)-Pt/La_{0.3}SrTiO₃/SrTiO₃. (a) AFM of La_{0.3}SrTiO₃/SrTiO₃ (b) Schematic illustration. (c) NLHE without a gate voltage and with a 0.4 gate voltage V_{TG} .

(b-2) Conversely, in reduced (112)-SrTiO_{3-x} annealed in vacuum, where oxygen vacancies are abundant but the overall symmetry remains essentially cubic, no NLHE is observed. The (112)-SrTiO₃ substrate was first etched and annealed under the same condition mentioned in the Method section of the main text. Subsequently, an amorphous Al₂O₃ layer was patterned on the surface of (112)-STO substrate, leaving the active Hall bar area exposed to the ambient conditions. The substrate was then transferred into the PLD chamber and annealed for 10 min in vacuum (oxygen pressure of 10⁻⁴ mbar) at 800 °C to create oxygen vacancies in the Hall bar channel. The sample was finally cooled down to room temperature under the same ambient pressure. In this configuration, the oxygen vacancies in the STO substrate contribute to the transport. However, the system experiences only weak symmetry breaking since the surface layer is not under the application of an electrical field. One may regard this situation as analogous to the (112)-Pt/La_{0.3}SrTiO₃/SrTiO₃ situation without gate voltage applied. As a result, this reduced (112)-SrTiO_{3-x} mainly retains the bulk cubic symmetry. Such a system is not expected to undergo any symmetry breaking and therefore does not exhibit NLHE, as clearly demonstrated in Fig. RR2.2,

Fig. RR2.2 Nonlinear Hall effect in reduced (112)-SrTiO_{3-x} crystal.

(c) Octahedral rotations:

In response to the Reviewer's example regarding the influence of octahedral rotations on the spin property of SrRuO₃, we performed controlled experiments by growing two SrRuO₃ thin films on (112)-SrTiO₃ substrates under different conditions. **Our main conclusion** is that despite variation in oxygen vacancy concentration and possible octahedral distortions, neither sample exhibits measurable NLHE and MOKE signals.

(c.1) Sample growth: *Sample #1*: (112)-SRO/STO with oxygen vacancies intentionally introduced in STO. Nominal 8 u.c. SRO was grown on (112)-STO at 800 °C under oxygen pressure of 10⁻⁴ mbar, mimicking the growth condition used for the 2DEG formation in (112)-LAO/STO. *Sample #2*: (112)-SRO/STO with minimised oxygen vacancy density. Nominal 8 u.c. SRO is grown on (112)-STO at 600 °C under an oxygen pressure of 0.13 mbar, thereby substantially suppressing the defect formation .

(c.2) NLHE and MOKE studies of (112)-SRO/STO

As shown in Fig. RR2.3, that neither sample displays sizable NLHE or MOKE signal. These results unambiguously rule out octahedral rotation (and/or zigzag, as will be discussed in Answer 2.2) as the primary mechanism responsible for the nonlinear transport and magneto-optical phenomena reported in our work.

Fig. RR2.3 NLHE and MOKE data acquired on two (112)-SRO/STO samples. (a) NLHE comparison.
(b) MOKE signal comparison

In summary, the fundamental origin of the observed effects is the symmetry breaking imposed by crystallographic orientation engineering. While defects may influence the transport properties, their role is secondary and derivative, as their impact itself arises from the underlying orientation-induced symmetry breaking.

Action 2.1: We have now added to the SI the results on (112)-SRO/STO. KTO results are the subject of a separate work and will not be disclosed at this moment.

Question 2.2. Interpretation of the zigzag pattern: The discussion of the zigzag pattern at the interface remains ambiguous. The authors conclude that such patterns do not generate physical effects, but no clear justification is given. Transport properties and spin are very sensitive to microscopic features. Furthermore, they cite Annadi et al. (Nat. Commun. 4, 1838 (2013)) to claim that zigzag features are also present in (110) interfaces. However, the zigzag reported in that work extends over only about one unit cell and is hardly visible in scanning transmission electron microscopy (STEM) images. In contrast, the zigzag modulation observed in the present (112) samples is much more pronounced, extending over several unit cells with a clear periodic composition modulation, as seen directly in STEM image (Fig. 1e). Indeed, when the contrast in Fig. 1e is enhanced, the zigzag pattern becomes very evident and is directly discernible in the STEM images. From this perspective, I find it inappropriate to consider the two cases equivalent. From this point of view, I disagree with the authors' statement: Both

images demonstrate a sharp interface between the thin film and the substrate (line 99 on page 4). Such roughness is unlikely to average out in a Hall bar geometry and may instead contribute a finite value to the measured transport. Additionally, the anisotropy of this roughness, as seen in Fig. 1e,f, could itself be a possible origin of the observed transport anisotropy. I strongly encourage the authors to examine this point more carefully.

Answer 2.2: First, we thank the reviewer for the careful inspection of the STEM images and for bringing the zigzag modulation of the interface into further discussion. After a careful assessment, we agree with the Reviewer that the zigzag modulations in (110) and (112) interfaces are different.

For the (110)-oriented SrTiO₃ interface/surface, two in-plane mirror planes exist (i.e., $M_{[\bar{1}10]}$ and $M_{[001]}$), which constrains the interface/surface zig-zag pattern to a straight step configuration, as illustrated in Fig. RR2.4a. In contrast, the (112)-oriented SrTiO₃ interface/surface retains only one in-plane mirror plane (i.e. $M_{[1\bar{1}0]}$), which might explain the more pronounced zigzag pattern observed in the STEM image of (112)-orientated substrate. This reduced symmetry enforces the zig-zag configuration at the interface/surface as schematically illustrated in Fig. RR2.4b, where even the step edge adopts a zig-zag pattern. Notably, such zig-zag pattern will become more evident in higher index orientation, such as the (001)-oriented SrTiO₃ with miscut 4° to [110], corresponding to the Miller index (1 1 20) [see: *Tailoring the domain structure of epitaxial BiFeO₃ thin films*, Current Opinion in Solid State and Materials Science 18 (2014) 39–45].

Fig. RR2.4 Zig-zag pattern analysis in (a) (110)-orientated, and (b) (112)-orientated surfaces.

Although we did not observe long-range, periodic zig-zag pattern in our (112)-LAO/STO sample by AFM, we acknowledge that such local structural modulations (if

present) may contribute to the observed effect. However, similar to the case of defects discussed above, these interfacial zigzag modulations are themselves a consequence of crystallographic orientation and act only in a derivative manner. This interpretation is supported by the control experiments on $\text{La}_{0.7}\text{Sr}_{0.3}\text{TiO}_3$ transistor, which exhibit clear nonlinear transport despite the absence of long-range zigzag patterns in AFM characterization (see Fig. RR2.1a). These results confirm that the zigzag interface modulations are not essential for the manifestation of nonlinear transport effects.

Action 2.2: To address this critical point, we have added the (112)-SRO/STO data in SI to state the effect of zigzag together with the illustration. Also, we have now revised the statement on the ‘interfacial sharpness’. We have added a section in the SI to discuss the interface zig-zag pattern.

Question 2.3. Definition of θ^{QWP} : Although the authors define θ^{QWP} as the angle between the fast axis of the QWP and the polarization of the incident light, this explanation is somewhat too brief. It would be helpful to provide a more explicit reference direction (e.g., relative to the crystal axes or the measurement geometry in the case of linearly polarized light) to avoid ambiguity.

Answer 2.3: We thank the reviewer for this suggestion. We define the angle of the quarter-wave plate, θ^{QWP} , as the angle between the fast axis of the QWP and the linear polarization direction of the incident beam, which is fixed along the laboratory axis x^{Lab} , as shown in Fig. RR2.5. The sample is oriented such that its [112] direction parallel to the light propagating direction, i.e., the z^{Lab} . We note that, the in-plane crystallographic axis of the sample are generally not aligned with the laboratory axis x^{Lab} or y^{Lab} , resulting in an angular offset, θ^{Off} . This angle offset θ^{Off} occurs as a shift in the measured and fitted curves of photocurrent vs QWP angle. To avoid any potential ambiguity, we have added a discussion of this effect in the SI.

Fig. RR2.5 Relationship between the QWP angle and particular crystallographic direction.

Action 2.3: We have now given a more detailed description of the θ^{QWP} in the Supporting Information with this figure presented.

Question 2.4. Fitting of the circular photogalvanic effect (CPGE)/ linear photogalvanic effect (LPGE) Data (Figs. 2b, 2c, and R2.4): The authors state that the reviewer "may have misunderstood" and that the fits are clearly good. However, I did not misunderstand; my point concerned the significant discrepancies between the experimental data and the total fit, especially near 90 degrees and 150 degrees in Fig. R2.4. In fact, in Wang et al., Phys. Rev. Lett. 128, 187401 (2022), the CPGE in (111) LAO/STO interfaces is fitted with excellent agreement across all angles and intensities. The substantial mismatch observed here suggests that additional factors may be at play. Moreover, in the new version of the manuscript, the separation into LPGE and CPGE components has been removed, which makes the text difficult to follow, and Figs. 2b and 2c appear to be interchanged. These figures need to be carefully re-presented. I am also concerned that the sum of the red (CPGE) and blue (LPGE) curves does not reproduce the black "total fit" curve in Fig. 3c (original version). For example, at around 135 degrees, the values of LPGE and the total fit differ, despite CPGE being nearly zero at this point. Is the fitting procedure correct? Has any angular offset been corrected for? If so, this should be clearly described in the manuscript.

Answer 2.4: First, we would like to thank the reviewer for pointing out the mistake.

Fig. 2b and Fig. 2c were in the wrong place.

(a) Regarding the experiment and fitting quality of the photocurrent.

We acknowledge the Reviewer's comment regarding the fitting of the CPGE data, and we agree that a discrepancy exists between the experimental data and the fitted curves. Such minor deviations may arise from light-polarization-dependent photo-thermoelectric effects or other polarization-dependent effects along the optical path that are not fully calibrated. Nevertheless, the overall agreement between the experiment and fit is sufficiently good to confirm the manifestation of the CPGE under perpendicular illumination and resolve the contrast between the two in-plane crystallographic directions. In Table RR2.1 we present the fitting parameters for the $[1\bar{1}0]$ direction, where the R^2 reaches 0.93 (with $R^2 = 1$ representing a perfect fit). This indicates that our fitting is quantitatively reliable.

Table RR2.1 Fitting parameters of CPGE in direction $[1\bar{1}0]$

	Abs. value	Standard error
CPGE component	0.372 nA	0.0197 nA
LPGE component	0.392 nA	0.0197 nA
θ^{off}	52.88°	0.669°
R^2	0.93	N/A

The reviewer cited the paper of Wang et al. [Phys. Rev. Lett. 128, 187401 (2022)], which indeed show an excellent fit in LAO/STO system compared to our results. However, it should be noted that even in that study, not all the experimental data exhibit a perfect agreement with the fits. For example, Fig. 4b of Wang et al., the data at around 100° and 360° deviate from the fitted curve despite an overall R^2 of about 0.97. Similar deviations can also be found in other published papers (see Fig. RR2.7): (a) Fig. 4c of [Duan et al., Nat. Nanotechnol. **18**, 867 (2023)] ($R^2 \approx 0.94$); (b) Fig. 5b of [Zhu et al., Nat. Commun. 13, 7702 (2022)] ($R^2 \approx 0.83$); (c) Fig. 2a of [Liu et al., Nat. Commun. 11, 323 (2020)] ($R^2 \approx 0.85$).

Fig. RR2.6 CPGE results in other papers. (a) Duan et al., Nat. Nanotechnol. **18**, 867 (2023) (under gate voltage of -50 V), (b) Zhu et al., Nat. Commun. 13, 7702 (2022), (c) Liu et al., Nat. Commun. 11, 323 (2020) (the red curve)

The main purpose of our CPGE study is to demonstrate the symmetry breaking, and in this respect, the fitting quality is fully adequate. The minor deviations noted above do not compromise the validity of our conclusions regarding CPGE and the associated symmetry considerations.

(b) Regarding the angular offset & the comparison between the total fit and the sum of LPGE and CPGE in direction $[11\bar{1}]$.

We thank the reviewer for the very careful examination of our results. Indeed, an error was present in the previous version of the manuscript in the fitting of the CPGE along $[11\bar{1}]$. Specifically the offset angle in LPGE and CPGE components was mistakenly entered as $+51.9^\circ$ to -51.9° . In Fig. RR2.7 we now present the corrected results for comparison. With this correction, the data are accurate and clearly represented.

Fig. RR2.7 CPGE study in direction $[11\bar{1}]$ with fit and the corrected CPGE and LPGE component.

Action 2.4: we have added a dedicated section in SI discussing to the possible deviation of the fitting process. The discussion of the offset angle θ^{off} has also been included in the SI for clarity. Furthermore, to provide a clearer understanding of the relationship between the individual CPGE and LPGE contributions and the total fit (for the $[1\bar{1}0]$ direction), we now present the two fitted components separately in SI.

Response to Reviewer #3:

Question 3: I carefully read the rebuttal of the authors to all three reviewers. I have an overall sense that the authors carefully addressed most of the concerns and took the necessary actions in revising the manuscript and in consequently expanding the content of the supplementary information. Thus, I recommend the publication of the present manuscript "Magnetization generation and giant nonlinear transport at symmetry-engineered interfaces via crystallographic orientation" by H.-B Zhang et al., in Nature communications. I believe it will be of great interest to many scientific communities working on nonlinear transport, and their theories, in various materials systems.

Answer 3: We really appreciate the reviewer's valuable suggestions to make this manuscript better.

Question 3.1: Regarding: Action 3.2.3 & 3.2.4: "We believe current manuscript is already a complete piece of work and we would like to study the gate tunability of this

system comprehensively in a separate work. We prefer to publish Fig. R3.4 separately." My comment: I appreciate the authors' effort and openness to disclose unpublished results, and agree that this is a substantial addition which warrants a detailed investigation for a follow-up publication.

Question 3.2: Regarding: "To further check the linear dependence in these ordinary Hall effects, we conducted the Hall measurement with the magnetic field applied as large as ± 8 T. As demonstrated in Fig. R3.3b, the linearity of these curves at all temperature ranges is maintained up to ± 8 T. Therefore, the magnetic field of ± 3 T is sufficient for the Hall measurement to investigate the carrier density and mobility." My comment: I respectfully disagree with the statement that the ordinary Hall effect is purely linear in the whole magnetic field range. It appears to me that the Hall effect is in fact slightly non-linear in field at the lowest temperatures. Overall I do not think this affects the conclusions of the manuscript, because this non-linearity is still quite small. However, the authors do not ignore that when multiband conduction and Hall effect is involved, the low-field and high-field regions may yield very different Hall coefficients, and that their relation to the carrier density of either or both subbands is not the same. In particular, only the high-field Hall coefficient can be taken as inversely proportional to the total carrier density in the system. I would not insist either on a two-bands fit of the Hall effect alone because it is known to be problematic due to the large number of free parameters in the model. I'm looking forward to a more detailed analysis in future studies of the Hall effect in these systems.

Answer 3.1 and 3.2: We highly value the reviewer's rigorous scientific attitude and profound knowledge. We plan to further optimize the growth condition to increase interface quality and have more detailed analysis in future studies of the electrical transport in these systems, including the Hall effect and diffusion of current-induced magnetization, etc.

Action 3.1 and 3.2: No action taken.

Response Letter Version 3

Reviewer #2 (Remarks to the Author):

I would like to thank the authors for their efforts in revising the manuscript in response to my comments. Also, I apologize for the wording mistake in my previous Question 2.1, where I wrote 'breaking' instead of 'preservation' of the mirror planes. The additional experiments and discussions, particularly regarding defects and octahedral rotations, convincingly support the symmetry-based interpretation. I am satisfied that my concerns have been fully addressed, and I recommend the manuscript for publication.

Answer: We thank all the reviewer for the valuable suggestion to make this manuscript better.

Action: No action taken.